# Unveiling the intellectual structure of informality: Insights from the socioeconomic literature

**Nelson Alfonso Gómez-Cruz**[1,2]*, **David Anzola**[1,3], **Aglaya Batz Liñeiro**[1]

1 Innovation Center, School of Management, Universidad del Rosario, Bogotá, Colombia, 2 School of Management, Colegio de Estudios Superiores de Administración (CESA), Bogotá, Colombia, 3 School of Sociology, University College Dublin, Dublin, Ireland

* nelson.gomez@urosario.edu.co

**Data Availability Statement:** All relevant data are within the paper and its Supporting Information files.

**Funding:** This research has been developed in the framework of the Colombia Cientifica – "Alianza EFI" Research Program, with code 60185 and

## Abstract

In the socioeconomic sphere, the concept of informality has been used to address issues pertaining to economic dynamics, institutions, work, poverty, settlements, the use of space, development, and sustainability, among others. This thematic range has given way to multiple discourses, definitions and approaches that mostly focus on a single phenomenon and conform to traditional disciplinary lines, making it difficult to fully understand informality and adequately inform policymaking. In this article, we carried out a multilevel co-word analysis with the purpose of unveiling the intellectual structure of socioeconomic informality. Co-occurring document keywords were used, initially, to delimit the scope of the socioeconomic dimension of informality (macro level) and, later, to identify its main concepts, themes (meso level) and sub-themes (micro level). Our results show that there is a corpus of research on socioeconomic informality that is sufficiently differentiable from other types of informality. This corpus, at the same time, can be divided into six major themes and 31 sub-themes related, more prominently, to the informal economy, informal settlements and informal institutions. Looking forward, the analysis suggests, an increasing focus on context and on the experience of multiple 'informalities' has the potential, on the one hand, to reveal links that help unify this historically fragmented corpus and, on the other hand, to give informality a meaning and identity that go beyond the traditional formal-informal dualism.

## 1. Introduction

Socioeconomic informality is a complex, multidimensional and context-dependent phenomenon in which social, economic, political, and geographical variables intervene [1]. The composition and impact of informality vary across economies and regions [2]. In emerging and developing countries, informal work accounts for around 30–35% of GDP and 70% of the total employment [3]. Globally, informal employment reaches 61.2% [4], which represents close to 2 billion people. Similarly, it is estimated that one in eight people in the world lives in informal settlements under precarious conditions [5]. In urban areas, informal settlements shelter 25%

contract number FP44842-220-2018, funded by The World Bank through the call Scientific Ecosystems, and managed by the Colombian Ministry of Science, Technology and Innovation. The funders had no role in study design, data collection and analysis, decision to publish, or preparation of the manuscript.

**Competing interests:** The authors have declared that no competing interests exist.

of the population [6]. Consequently, informality has profound implications for global development and sustainability.

Research on informality has been characterized by its compartmentalized development, with a predominant focus on specific themes such as informal settlements, the informal economy, urban informality, the informal labor market, informal employment, informal institutions, and others [1, 7]. Unfortunately, this fragmented approach has resulted in a lack of interconnectedness between these themes, obscuring their potential linkages at the core of various socioeconomic issues. As a result, several authors have attempted to develop taxonomies or classifications of informality, aiming to improve the understanding of the primary factors contributing to it. However, these efforts have arisen from diverse thematic perspectives.

For example, it is common to find general analyzes on the informal economy [1, 2] or on informal settlements [8]. Informality, however, has consistently been approached from a singular, narrowly framed perspective. Boanada-Fuchs and Boanada [7] concentrated their study on publications related to 'informal economy', 'housing', 'land tenure' and 'urban planning', while Venerandi and Mottelson [9] proposed a taxonomy exclusively focused on categorizing 'informal settlements'. Furthermore, Dovey et al. [10] acknowledge that certain patterns lead to the formation of informal settlements and influence the emergent urban design or informal morphogenesis. Alsayyad [11] discusses various patterns of urban informality, such as invasion, survival, assimilation, adaptation, and cooperation. Roy [12] delves into the epistemology of public policies employed in urban planning and presents a characterization of these policies. Although most approaches have traditionally centered on the topic of informal settlements, Fernandez and Villar [13] put forward a taxonomy to better understand the informal labor market by analyzing three characteristics: choice, productivity, and barriers. However, efforts to establish general frameworks are scarce and are often focused on ad hoc definitions or on dimensions that fall within the domain of expertise of the authors [1, 7].

This lack of thematic unification makes it difficult to (i) clarify informality's thematic boundaries, dimensions, and scope, (ii) achieve a basic conceptual understanding, (iii) offer satisfactory measurement alternatives and, in general, (iv) sufficiently integrate the research areas for which the concept is important. Hence, this paper endeavors to address the question: what is the intellectual structure of informality research?

To address this question, we employed a multilevel variant of co-word analysis [14], which enables us to minimize bias in the development and classification of topics addressed in the literature. By using this approach, we aim to reveal the central research problems related to socioeconomic informality. Co-word analysis is a quantitative content analysis technique [15, 16] that allows mapping the intellectual structure of a research field through the relationships between concepts (co-occurrences) within a corpus of literature. In this text, we use the method at three different levels of detail to (i) determine the frontiers of research on socioeconomic informality and its differences with other forms of informality (macro level), (ii) identify the main concepts, themes (meso level) and sub-themes (micro level) addressed in informality studies, (iii) explore the relationships between the various sub-themes and (iv) assess, from a general perspective, the relative importance and level of development of the various sub-themes.

To achieve this goal, the paper is organized as follows. The Methodology section provides an overview of the methodological framework. The Results section presents the main outcomes of the multilevel co-word analysis. Initially, it establishes the boundaries of socioeconomic informality (macro level). Subsequently, its main themes and sub-themes are identified and characterized (meso and micro levels). The Discussion section introduces a taxonomy that succinctly captures the intellectual structure of socioeconomic informality and delves into its evolution and future opportunities. Lastly, final remarks are provided in the Conclusions section.

## 2. Methodology

This research employs a multilevel co-word analysis to identify and analyze the intellectual structure of socioeconomic informality. Co-word analysis is a content analysis and scientific mapping technique [15, 17] based on the co-occurrence of concepts found in the scientific literature. This approach allows to (re)construct and visualize, in a comprehensive and systematic way, the intellectual structure of a field as a weighted network of interrelated concepts. In this article, the concepts were extracted from the 'author keywords' of the literature indexed in Scopus. Those concepts that appear close within the network usually have a thematic relationship [17]. Therefore, a set of concepts that are densely connected to each other, but loosely connected to other concepts in the network, form a cluster that, in turn, can be interpreted as a research theme. The relevance and level of development of each theme can be captured through network metrics in a strategic diagram.

Co-word analysis is used in this research on three levels. At the macro level, the general literature on informality is mapped to avoid bias on the delimitation of socioeconomic informality. The meso level extracts, in an explicit way, the literature on socioeconomic informality, characterizes it statistically and identifies its main research themes (meso clusters). At the micro level, each theme is divided into sub-themes (micro clusters). The degree of development and the importance of each sub-theme within its theme, that is, its relative strategic position, is analyzed in the section 4 Discussion. Fig 1 summarizes the methodological approach used.

### 2.1. Data collection

This study used academic literature on informality indexed in Elsevier Scopus, one of the largest curated scientific databases in the world [18]. Scopus data has been extensively used in bibliometric studies and co-word analysis [19, 20]. This database was chosen over alternatives such as Web of Science (WoS) or Google Scholar due to its coverage and academic rigor. Scopus has a broader coverage than WoS, especially in social sciences [21]. Recent research [22] indicates that 99.11% of journals indexed in WoS are also included in Scopus. In contrast, only 33.93% of journals indexed in Scopus are found in WoS.

Google Scholar, in turn, has broader coverage than Scopus, but faces significant issues regarding transparency and reproducibility [23, 24]. Firstly, it lacks a curated catalog, allowing the inclusion of non-peer-reviewed literature. Furthermore, search results are not consistently reproducible, and it does not support the use of boolean operators. Consequently, thanks to its balance between coverage and rigor, Scopus ensures that our analysis encompasses a broad spectrum of relevant literature (containing most of the WoS content) without compromising on the quality of data.

Two search strategies were considered: a general one (macro) and one restricted to the socioeconomic domain (meso). Both searches employed the 'title' and 'author keywords' fields. The strategy employed at the macro level captures the various uses of the term 'informal' and its variations (informality, informalization), according to the following statement:

$$\text{(TITLE (informal*) OR AUTHKEY (informal*))} \tag{1}$$

29,546 documents were retrieved, of which 21,463 contained author keywords (S1 File). The latter were used in the macro co-word analysis (see subsection 3.1 Macro level: A general view of informality research).

After analyzing this first set of data and its co-occurrence network, the keywords that (i) belonged to the socioeconomic domain, (ii) had 50 or more occurrences, and (iii) included the term informal or informality, were included in the construction of the second search strategy. The resulting statement was:

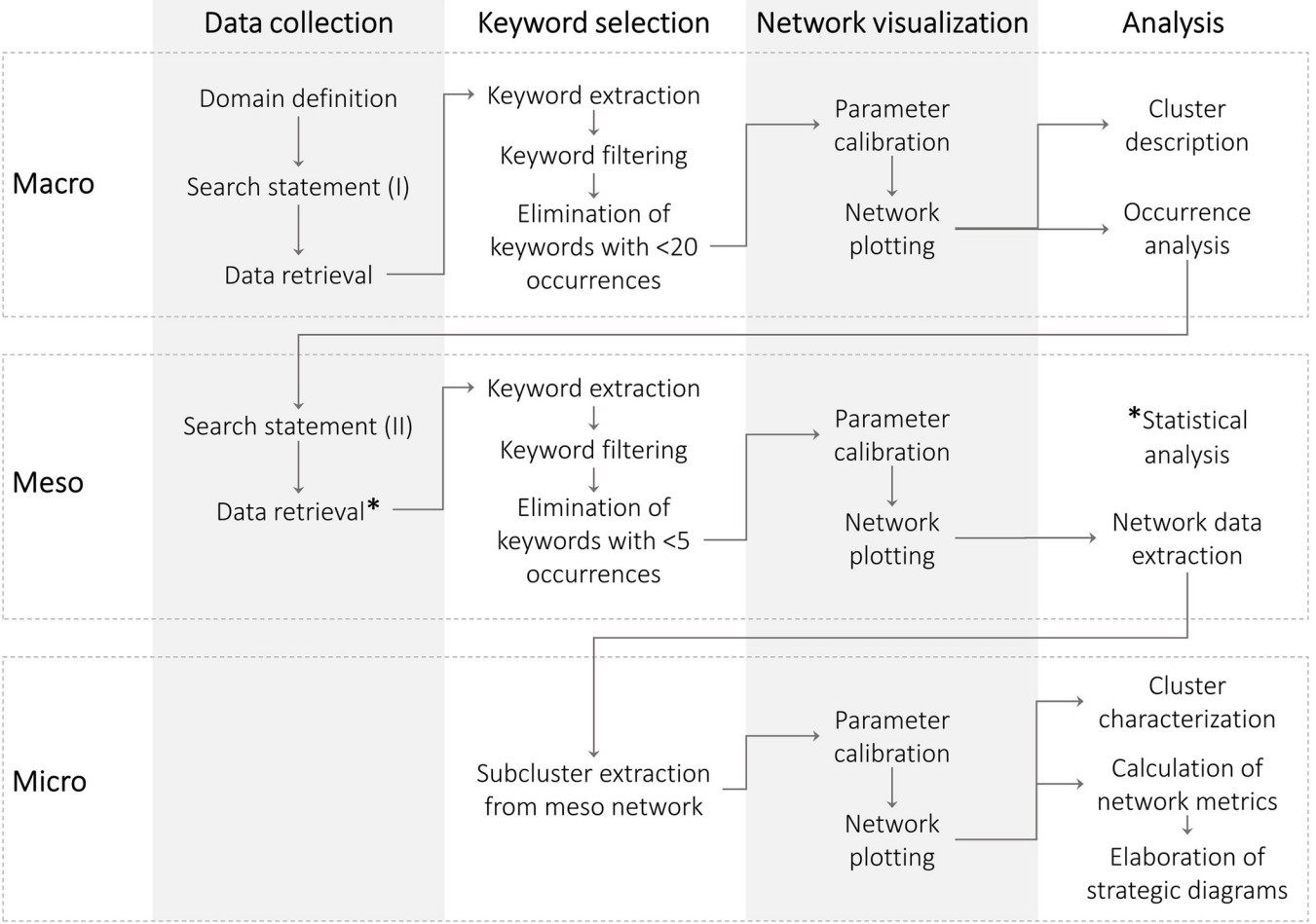

**Fig 1. Methodological framework.**

(TITLE (informality OR "informal econom*" OR "informal settlement*" OR "informal sector*" OR "informal sector*" OR "informal institution*" OR "informal employment" OR "informal network*" OR "informal work*" OR "informal social control" OR "informal payment*" OR "informal market*" OR "informal housing" OR "informal recycling" OR "informal control" OR "informal labor" OR "informal finance" OR "informal governance" OR "informal entrepreneur*" OR "informal trade") OR AUTHKEY (informality OR "informal econom*" OR "informal settlement*" OR "informal sector*" OR "informal institution*" OR "informal employment" OR "informal network*" OR "informal work*" OR "informal social control" OR "informal payment*" OR "informal market*" OR "informal housing" OR "informal recycling" OR "informal control" OR "informal labor" OR "informal finance" OR "informal governance" OR "informal entrepreneur*" OR "informal trade")) (2)

From this search, 10,236 documents were obtained and used for the statistical analysis of the literature on socioeconomic informality (see S2 File and subsection 3.2.1 Statistical analysis of literature). Of these, only 7,814 documents that contained keywords were used to create the co-word network presented in subsection 3.2.2 Co-word network of socioeconomic informality. Both searches were carried out on 17/11/2021 and included all document types available in Scopus. The records obtained for each search were stored in a CSV file for later analysis.

The networks elaborated at the micro level were developed from the clusters identified at the meso level, so they were indirectly based on the same data set retrieved with search statement (2).

## 2.2. Keyword selection

A thesaurus (controlled vocabulary) was built to filter each data set (macro and meso), according to the following rules [25]: (i) standardization of singular and plural words (e.g., settlement into settlements); (ii) combination of equivalent expressions in American and British English (e.g., urbanization into urbanization); (iii) integration of acronyms with their corresponding full names (e.g., non-governmental organizations into ngos); (iv) normalization of hyphenated expressions (e.g., neoliberalism into neo-liberalism); (v) reduction of synonymous expressions (e.g., small enterprise into small firm); (vi) stemming of derived words in justified cases (e.g., flooding into flood), and (vii) elimination of references to the territory (e.g., Latin America, India), research methodologies (e.g., case study, ethnography), and irrelevant words (e.g. informality, informal).

For the macro data set, 45,308 keywords were found in the raw data. After preprocessing, 44,874 unique terms were retrieved. Last, 496 terms that had 20 or more occurrences were included in the macro network. At the meso level, meanwhile, 17,005 keywords were extracted from the raw data. After the application of the thesaurus, 16,613 unique terms were found. Finally, 759 terms that had 5 or more occurrences were included in the meso network. The networks analyzed at the micro level were extracted directly from the meso level network data.

## 2.3. Network visualization

Once the keywords were preprocessed, the co-occurrence matrices for the macro and meso levels were calculated and mapped using the VOSviewer software [26]. In a co-occurrence network, the nodes represent the keywords and the edges the co-occurrences between them. The number of occurrences of a keyword within the corpus is represented by the size of its node. The larger a node is, the more frequent the keyword it represents. The thickness of an edge, for its part, indicates the number of documents in which the two connected keywords appear together. Once the network is built, the colors represent the thematic affinity between groups of nodes (clusters). Each cluster, therefore, can be explained from the nodes and edges it comprises.

Co-occurrence networks at the macro, meso, and micro levels were mapped using the Lin-Log normalization algorithm [27] included in VOSviewer. The attraction and repulsion parameters for the macro and meso networks were set to 2 and 0, respectively. For the micro-level networks, in contrast, different attraction and repulsion values were used according to their specificities. Clusters detection was performed with the clustering technique that is incorporated by default in VOSviewer, which is a variant of the modularity-based clustering methods developed in physics [28]. The VOS clustering technique includes a resolution parameter that determines the number of partitions that the algorithm will produce. The higher its value, the higher the number of clusters generated. The resolution of the macro network was set to 0.80 to convincingly highlight the socioeconomic cluster. In all other cases, this parameter was set to its natural value of 1.00. VOS viewer, unlike other tools, uses a unified paradigm that integrates normalization (visualization) and clusterization techniques [29]. For this reason, the parameters must be calibrated iteratively.

Finally, the data corresponding to each meso cluster were extracted to build sub-theme networks at the micro level. Each network, therefore, was clustered in such a way that it was

possible to obtain a set of relevant themes and sub-themes for research on socioeconomic informality.

## 2.4. Analysis

The co-occurrence network at the macro level was characterized in a general way, with the purpose of establishing clear demarcation criteria between socioeconomic informality and other forms of informality. A count of the occurrences of the socioeconomic informality keywords was performed to construct search statement (2). The meso-level corpus of literature, in turn, was statistically described and the network data were extracted for the purpose of reconstructing micro-level networks.

At the micro level, the networks were described from their structure and from the review of documents (for example, highly cited) contained in the corpus. For each of the networks at this level, a strategic diagram that contrasts the network metrics of density and centrality of the micro-clusters was constructed (see section 4 Discussion). For a given micro cluster, the density captures its level of internal coherence, while the centrality indicates its importance within the meso cluster to which it belongs. According to Pourhatami et al. [30], the density and centrality of a cluster can be calculated using the following equations:

$$D_L = \frac{2E}{N(N-1)}$$

where $D_L$ represents the density of the cluster $L$, while $N$ and $E$ indicate the total number of nodes and edges in $L$, respectively.

$$C_L = \sum_{i \in L} \sum_{j \in M} w_{ij} e_{ij}$$

where $C_L$ is the centrality of cluster L and M is the set of other clusters that make up the network in the micro level. $i$ denotes the nodes of $L$ and $j$ the nodes belonging to the clusters of $M$. $e_{ij}$ can take the value of 0 if there is no edge between $i$ and $j$, and 1 otherwise. Finally, $w_{ij}$ indicates the weight of the edge between $i$ and $j$.

A strategic diagram (Fig 2) allows the clusters of a network to be classified into four quadrants, according to their relative values of density and centrality. Within the diagram, the vertical axis represents centrality and the horizontal axis density. Its origin will be given by the average of the centralities and densities of the clusters that make up the network under analysis. The importance and maturity of each cluster will be defined according to the quadrant to which they belong. Clusters in quadrant I are relevant to the network and have a high level of internal cohesion (mainstream). Clusters in quadrant 2 have a high level of internal cohesion, but are isolated (ivory tower). Quadrant III includes underdeveloped and isolated clusters (chaos/unstructured). These clusters may, depending on the case, be emerging or in decline. The clusters in quadrant IV generally involve important concepts for the network that, however, are not sufficiently cohesive among themselves (bandwagon).

## 3. Results

### 3.1. Macro level: A general view of informality research

Since the literature on informality is both vast and severely fragmented, a two-level search strategy was adopted, first, to identify its nature and scope and, second, to avoid introducing an undesired bias in the corpus retrieval process. Fig 3 shows the macro network generated from the documents retrieved with search statement (1). Three major well-differentiated

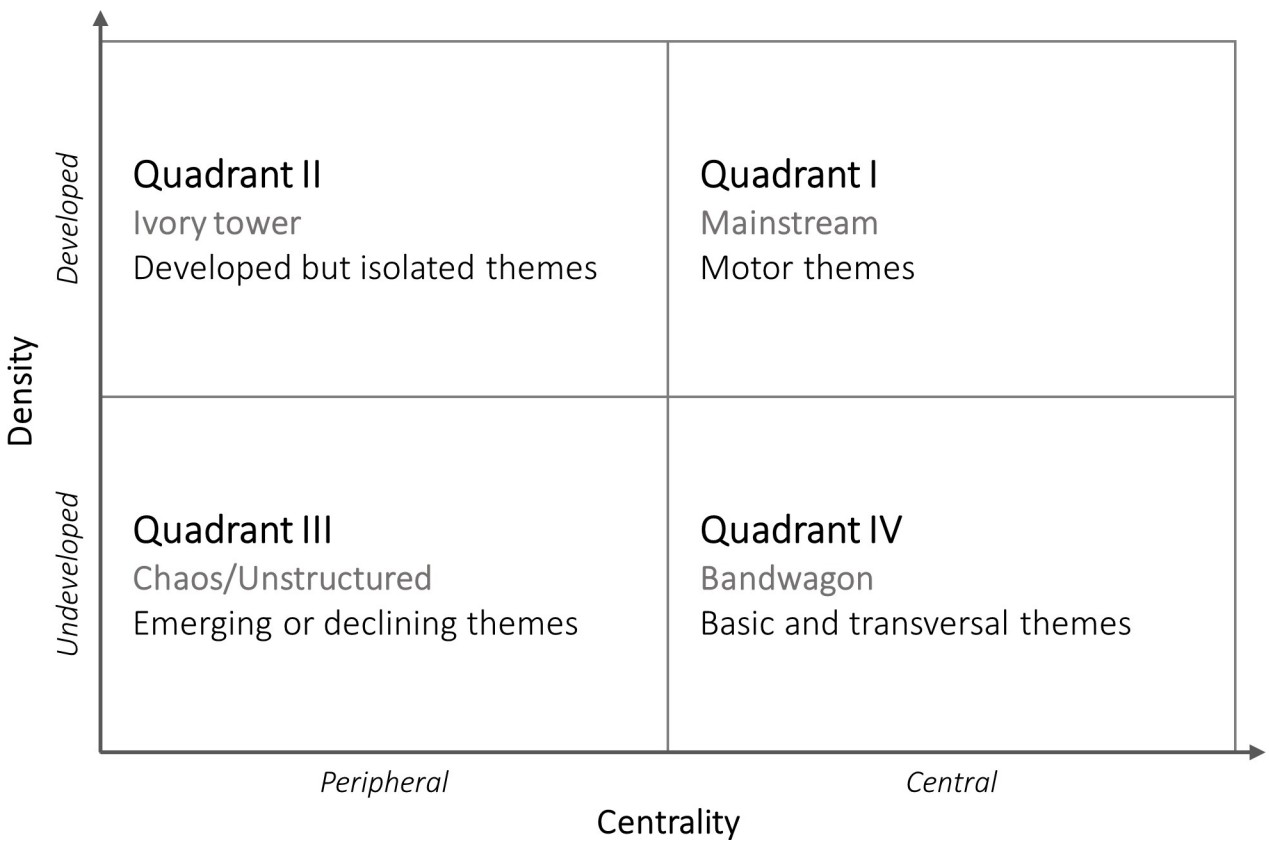

**Fig 2. Strategic diagram structure.**

clusters can be observed. The cluster in the upper part of the Fig 3 focuses on *Informal learning*. This topic has gained increased popularity in the literature and accounts for the largest portion of the corpus. The 'informal learning' node, in turn, is the largest in the network. Regularly, informal learning is defined in opposition to formal learning, that is, as any kind of learning occurring outside the formal education system [31] (a portion of the literature, however, centers on informal learning as the non-curricular learning that takes place in formal education environments (e.g., [32])). Thus, the cluster, overall, includes several nodes referring to a multiplicity of tools (e.g., 'mobile learning', 'social media', 'web 2.0'), mechanisms (e.g., 'experiential/lifelong learning', 'professional development') contexts (e.g., 'workplace/family learning', 'museums'), activities (e.g., 'serious games', 'mentoring', 'training') and agents ('children', 'adults', 'community of practice', 'online communities') that intervene in the diverse learning processes regularly labelled as *informal*.

A second cluster on *Informal care* is observed in the lower right corner. Informal care is generally described as the provision of unpaid care and assistance to someone in need. Often, the sick and the elderly are the recipients of care, while friends and family members are the providers [33]. Even though the largest nodes in the cluster: 'informal care', 'caregiver' and 'informal carer', are closely related conceptually, some disciplinary differences can be observed at lower-level nodes and micro clusters. A large amount of literature in the health sciences addresses the health condition of caregivers (e.g., 'burnout', 'caregiver stress') and those cared for (e.g., 'dementia', 'alzheimer'), and on the caregiving activity itself (e.g., 'nursing', 'quality of care'). In turn, a portion of the literature tackles informal care from a health economics

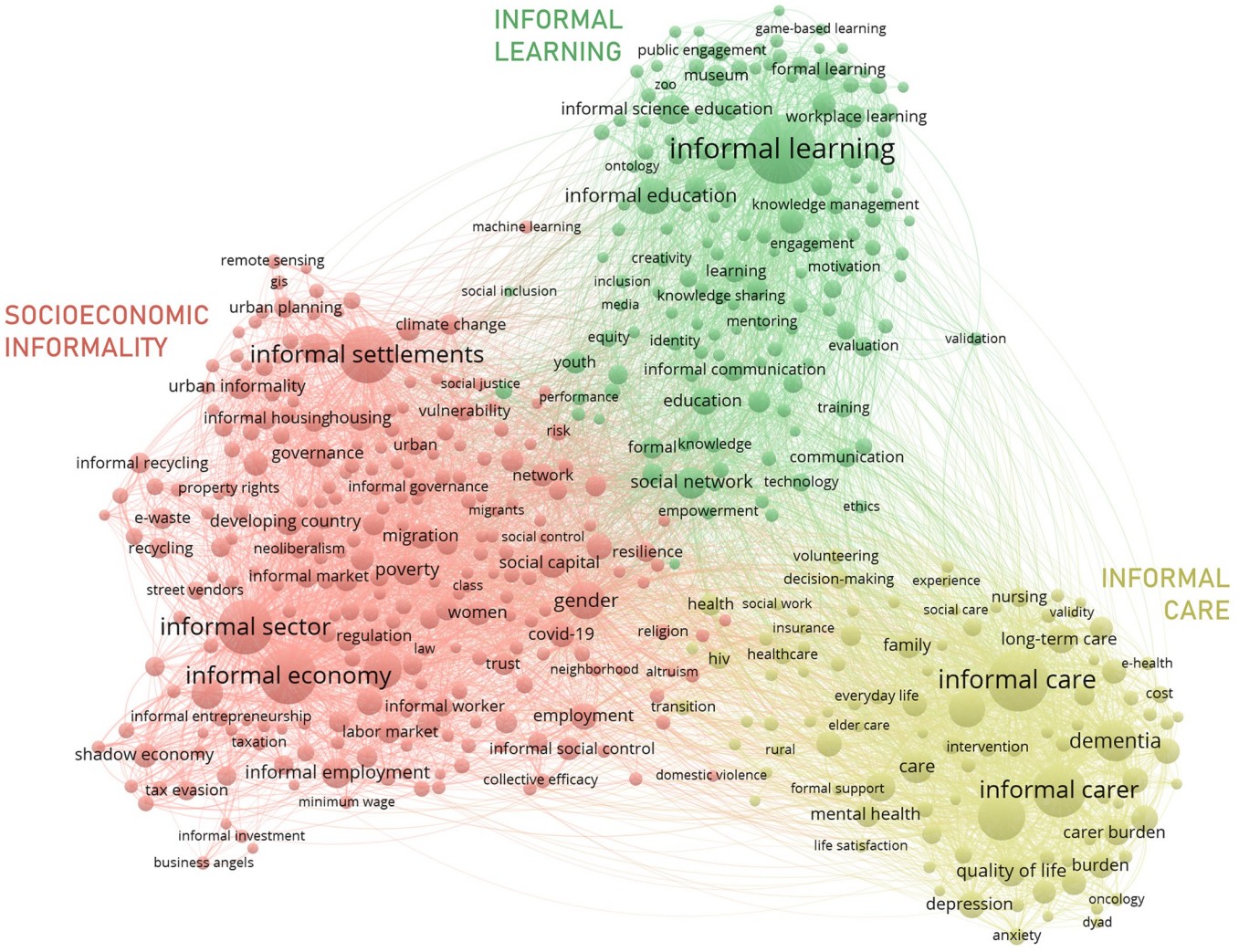

**Fig 3. Co-word network of informality research.**

perspective. This literature centers primarily on the valuation of this type of caregiving (e.g., 'economic value', 'economic evaluation', 'cost of illness'), as well as on its institutional impact (e.g., 'policy', 'housing'). Finally, some social science literature transversally centers on the diverse social conditions of informal caregiving, such as the type of social relationships that allow for unpaid work (e.g., 'social capital') or the predominant gendered nature of informal care (e.g., 'woman').

The final cluster, in the lower left corner, unlike the other two, does not display a unifying theme. It includes large nodes and sub-clusters with typical keywords for informality in the economy, the urban sector and varied institutional settings. This cluster is, undoubtedly, the one that more thoroughly accommodates research on *socioeconomic informality*. Since this review is primarily motivated by the desire to map the intellectual structure of the literature on the socioeconomic dimension of informality, the following sections will center exclusively on this third cluster. Although informal learning and informal care involve some decidedly relevant socioeconomic dynamics, the exclusion of these clusters from the co-word analysis seems warranted. First, the three clusters are clearly separated in the macro network, which suggests

the literatures are thematically distinct. Second, several nodes in the frontier between clusters e.g., 'community', 'education', 'technology' or 'social network', are sufficiently general to accommodate disciplinarily different discussions. Social networks, for example, can be used as a digital tool for informal learning [34], as a multifaceted mechanism of job mobility [35] or as a source of support for someone in need of informal care [36]. Third, there are variations in the connotation of "informality". Unlike the socioeconomic literature, most uses of "informal" in the learning cluster (e.g., as a prefix for learning, education, communication, reasoning) are either positively valued or simply acknowledged as qualitative different (without value judgements mediating the formal-informal distinction). Likewise, in many of these uses, "informal" and "formal" are not seen as opposites or mutually exclusive, a view that has historically shaped the socioeconomic literature [37–39].

It is worth pointing out that narrowing the analysis does not necessarily mean excluding the literature that loads onto the other clusters. Since a different search statement (2) is used to retrieve documents on socioeconomic informality, i.e., the meso level, some keywords that belong to the other macro clusters, and their corresponding literature, are equally retrieved and incorporated into the analysis. Understandably, however, they might have less relevance than in the macro network.

### 3.2. Meso level: Socioeconomic informality and their themes

**3.2.1. Statistical analysis of literature.** Fig 4 illustrates the evolution of the socioeconomic literature on informality since 1978 (previous years were not considered, as the volume of documents does not exceed 4 per year). Even though several foundational publications can be traced back to the mid-twentieth century [40], most documents retrieved were published after

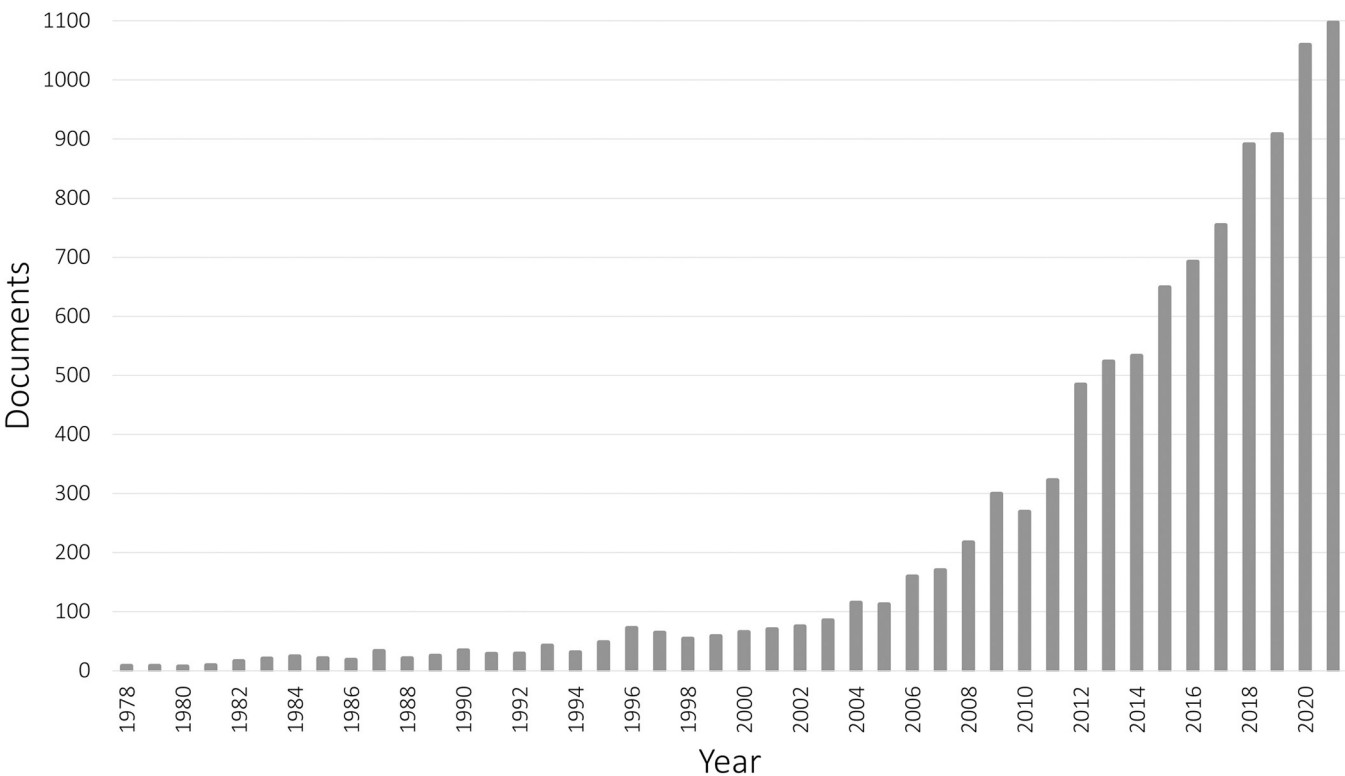

**Fig 4. Documents on socioeconomic informality published per year.** Source: Own elaboration based on Scopus data.

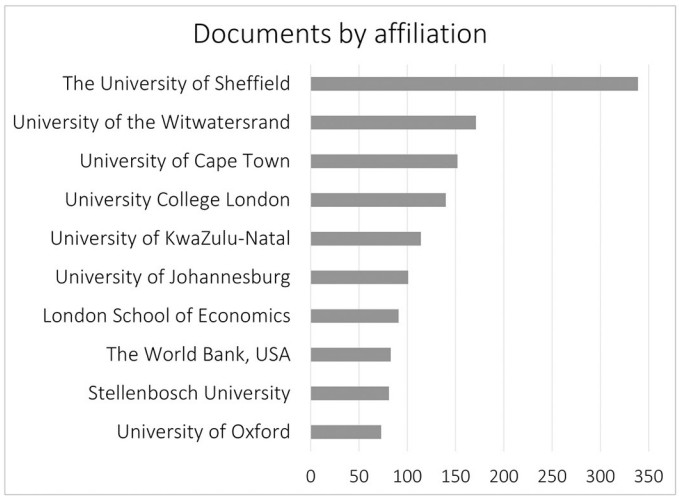

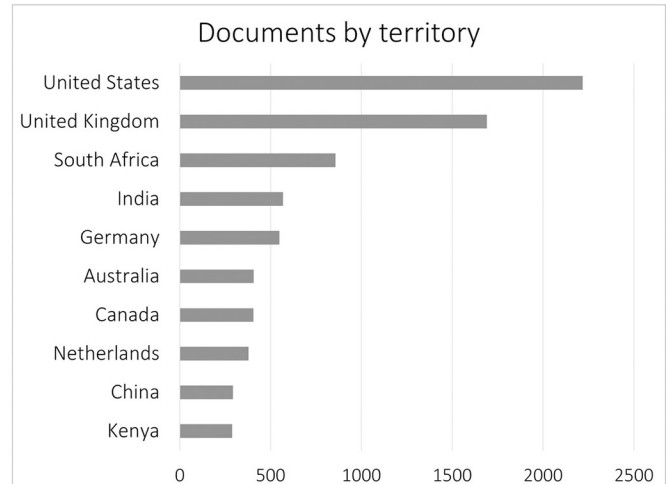

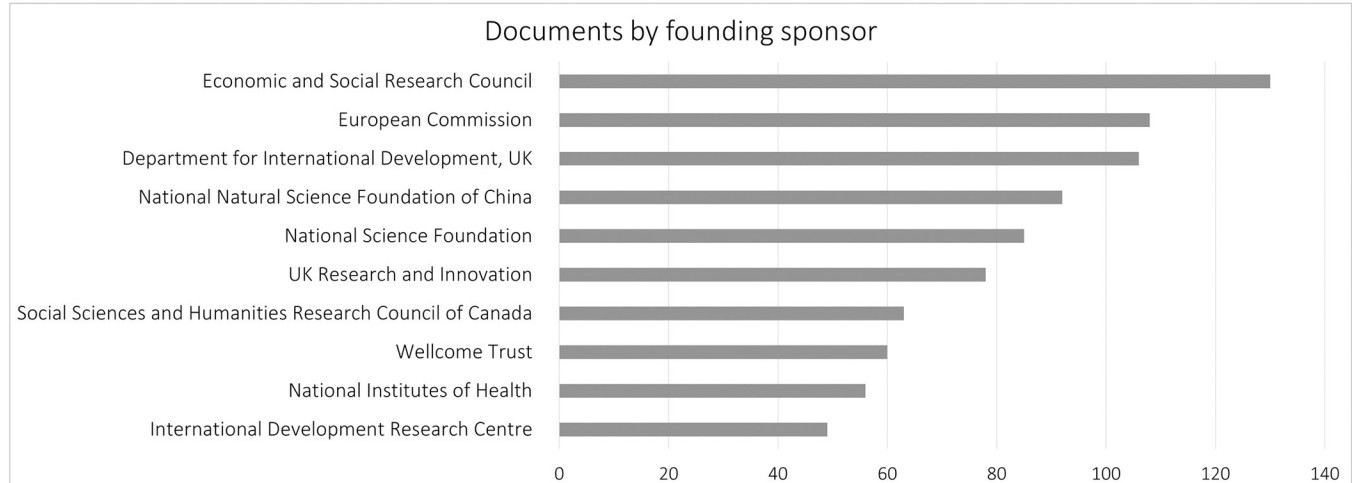

**Fig 5. Documents by affiliation, by territory and by founding sponsor in socioeconomic literature.** Source: Own elaboration based on Scopus data.

the nineties, and an important inflection point can be observed between 2005–2009. Over 93% of the documents in the corpus were published in the last 20 years.

Of the more than 10.000 articles retrieved using search statement (2), approximately 48% are produced by European researchers and institutions (Fig 5). While informality is more prevalent in emergent and developing countries [41], with the exception of South Africa, researchers from these countries only peripherally contribute to the corpus—although their contribution is increasing. Some differences are noticeable between and within regions, as well. African researchers participate in about 19% of the documents retrieved, with South Africa and Kenya standing out in this region. Likewise, Asian researchers contribute to about 21% of the corpus, with India and China leading in reported instances. Lastly, Latin America and the Caribbean is the region with the fewest publications, accounting for only 9% of the corpus, with Brazil and Colombia driving research in this region.

Table 1 lists the leading journals for publications on socioeconomic informality. Two major themes are readily identified: urban informality and economics and development. As it will be discussed below, these two themes are major drivers of the academic interest in informality. It

**Table 1. Leading journals that publish studies related to socioeconomic informality.** Source: Scopus and Scimago.

| Journal | Quartile | H-index | Freq. | % |
|---|---|---|---|---|
| World Development | Q1 | 175 | 111 | 1,28 |
| Environment and Urbanization | Q1 | 73 | 106 | 1,23 |
| Habitat International | Q1 | 78 | 105 | 1,22 |
| Urban Studies | Q1 | 147 | 101 | 1,17 |
| Sustainability | Q1 | 85 | 79 | 0,91 |
| International Journal of Urban and Regional Research | Q1 | 114 | 72 | 0,83 |
| Urban Forum | Q2 | 35 | 67 | 0,78 |
| Cities | Q1 | 90 | 65 | 0,75 |
| Indian Journal of Labour Economics | Q3 | 13 | 64 | 0,74 |
| International Journal of Sociology and Social Policy | Q2 | 39 | 63 | 0,73 |
| Journal of Developmental Entrepreneurship | Q3 | 25 | 57 | 0,66 |
| Journal of Development Economics | Q1 | 142 | 53 | 0,61 |
| Development Southern Africa | Q2 | 41 | 45 | 0,52 |
| International Development Planning Review | Q1 | 32 | 45 | 0,52 |
| International Journal of Environmental Research and Public Health | Q2 | 113 | 42 | 0,49 |

is worth noting that, while the contribution of researchers in emerging and developing economies is smaller, two regional journals make this list.

**3.2.2. Co-word network of socioeconomic informality.** Table 2 lists the keywords with the highest number of occurrences in the meso network. Interestingly, the six top-ranked terms include the prefix "informal". While the frequency of some keyword might be favored by the terms that were included in the search statement (2), it is clear that the current understanding of informality, and the terminology used to account for it, has been historically shaped by the formal-informal dichotomy. The table also evidences the influence of the economic, institutional, and spatial dimensions of informality. Several terms included, many of them with significantly high frequency, are decidedly linked to the diverse dynamics of informal markets, institutions, and settlements. Complementarily, some other keywords e.g., 'gender', 'informal network' or 'social capital', are not thematically narrow, but prove equally

**Table 2. Highly frequent keywords in 7814 documents.**

| No. | Keyword | OCC | % | No. | Keyword | OCC | % |
|---|---|---|---|---|---|---|---|
| 1 | informal economy | 1131 | 6,35 | 16 | informal social control | 124 | 0,70 |
| 2 | informal settlements | 1038 | 5,83 | 17 | Urbanization | 121 | 0,68 |
| 3 | informal sector | 1018 | 5,72 | 18 | Housing | 116 | 0,65 |
| 4 | informal institutions | 412 | 2,31 | 19 | developing country | 113 | 0,63 |
| 5 | informal employment | 240 | 1,35 | 20 | informal payments | 113 | 0,63 |
| 6 | informal work | 199 | 1,12 | 21 | informal housing | 106 | 0,60 |
| 7 | entrepreneurship | 197 | 1,11 | 22 | labor market | 106 | 0,60 |
| 8 | slum | 197 | 1,11 | 23 | informal market | 104 | 0,58 |
| 9 | gender | 195 | 1,10 | 24 | covid-19 | 102 | 0,57 |
| 10 | shadow economy | 169 | 0,95 | 25 | Employment | 101 | 0,57 |
| 11 | poverty | 158 | 0,89 | 26 | informal workers | 99 | 0,56 |
| 12 | informal networks | 151 | 0,85 | 27 | Migration | 96 | 0,54 |
| 13 | urban informality | 141 | 0,79 | 28 | formal institutions | 94 | 0,53 |
| 14 | governance | 130 | 0,73 | 29 | social capital | 93 | 0,52 |
| 15 | corruption | 127 | 0,71 | 30 | Development | 90 | 0,51 |

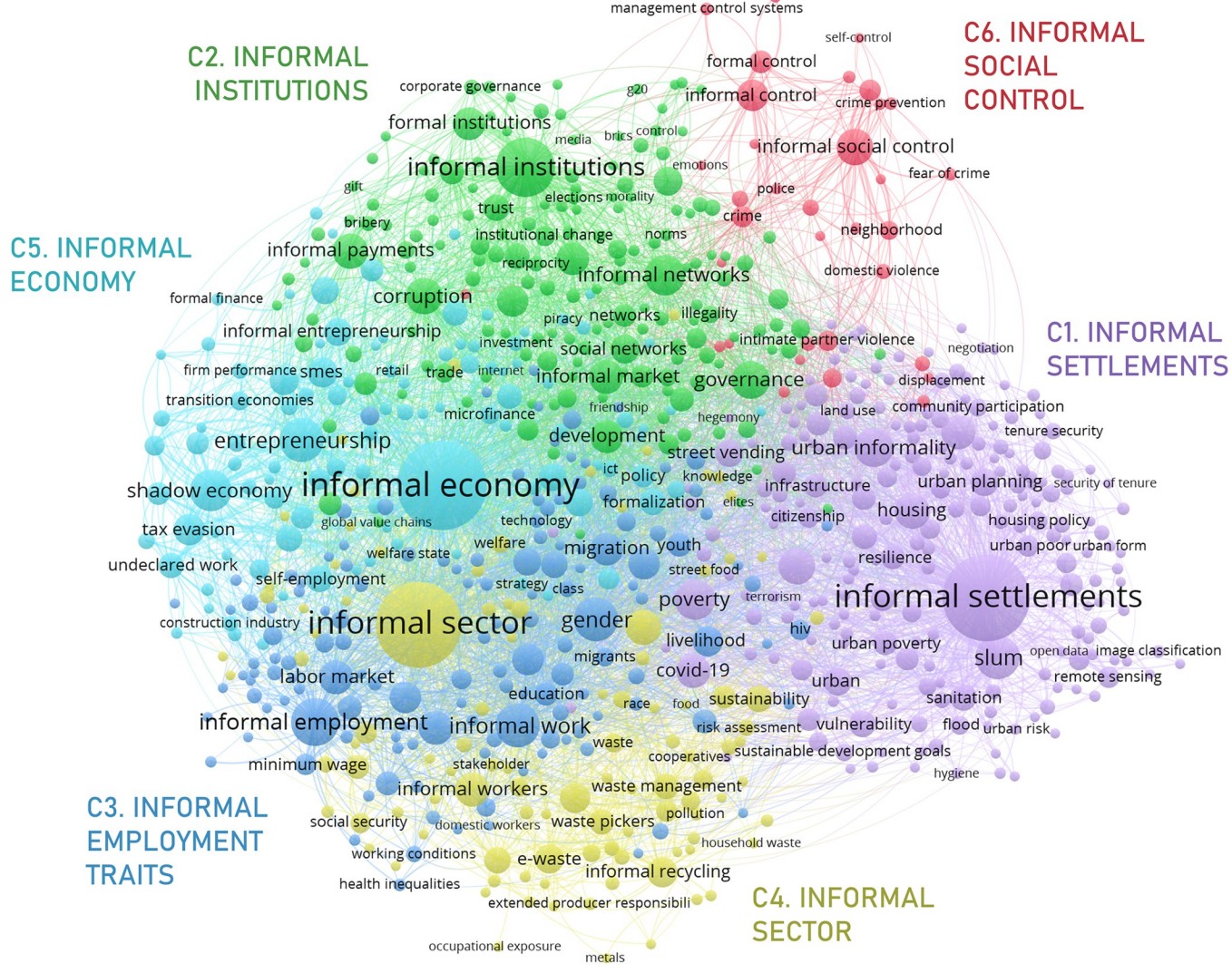

**Fig 6. Socioeconomic co-word network.**

important, for they, on the one hand, are transversal to the multiple dimensions of informality and, on the other hand, link or bring together distinctive processes and mechanisms of these multiple dimensions. Finally, it is worth pointing out that Covid-19 made it to the list at number 24. The pandemic has significantly affected informal populations and poses major challenges to recovery in the upcoming years [42]. Despite its relative novelty, it is necessary to be mindful of its potential impact on informality research moving forward.

The meso level network, centering on socioeconomic informality proper, is shown in Fig 6. The relevance of the economic, institutional, and spatial dimensions of informality is more evidently depicted by the fragmentation of the meso network. Specialized clusters are found for each dimension (although the economic aspects of informality are spread among three different clusters with somewhat different focus: C3-5). Overall, apart from the "informal social control" meso cluster, it is a highly integrated network: nodes are close together (edge length is a factor of thematic distance) and cluster boundaries overlap. There are, however, some

distinctive dynamics at the level of micro clusters that will be explored in more detail in the following subsections.

### 3.3. Micro level: Subthemes in socioeconomic informality

**3.3.1. Informal settlements.** The "informal settlements" meso cluster, C1 (Fig 7) is the largest in the meso network. It prominently captures the spatial aspects of informality, especially those related to the emergence and dynamics of informal settlements and slums [43, 44], informal housing practices [45], and the use of public space in the urban informal economy [46]. The cluster contains 6 micro clusters: C1-1 *urban informality*, C1-2 *informal housing*, C1-3 *urban planning & monitoring*, C1-4 *urbanization*, C1-5 *vulnerabilities*, and C1-6 urban sustainability.

C1-1 captures, in a general way, the reciprocal influence between public management ('urban governance', 'urban policy', 'planning',) and the dynamics of informality in the urban space ('street vending', 'appropriation', 'informal transport') (e.g., [47, 48]). Particularly, it addresses, on the one hand, problems that arise from this relationship ('eviction', 'displacement', 'gentrification', 'squatting') [49–51], and, on the other hand, struggles faced by 'citizens' in their search for their 'right to the city' ('social movements', 'self-governance') [52, 53].

C1-2 analyzes the problem of '[informal] housing', its roots in 'urban poverty' and how it thwarts 'urban development' (e.g., [54–56]). The cluster includes nodes referring to the various

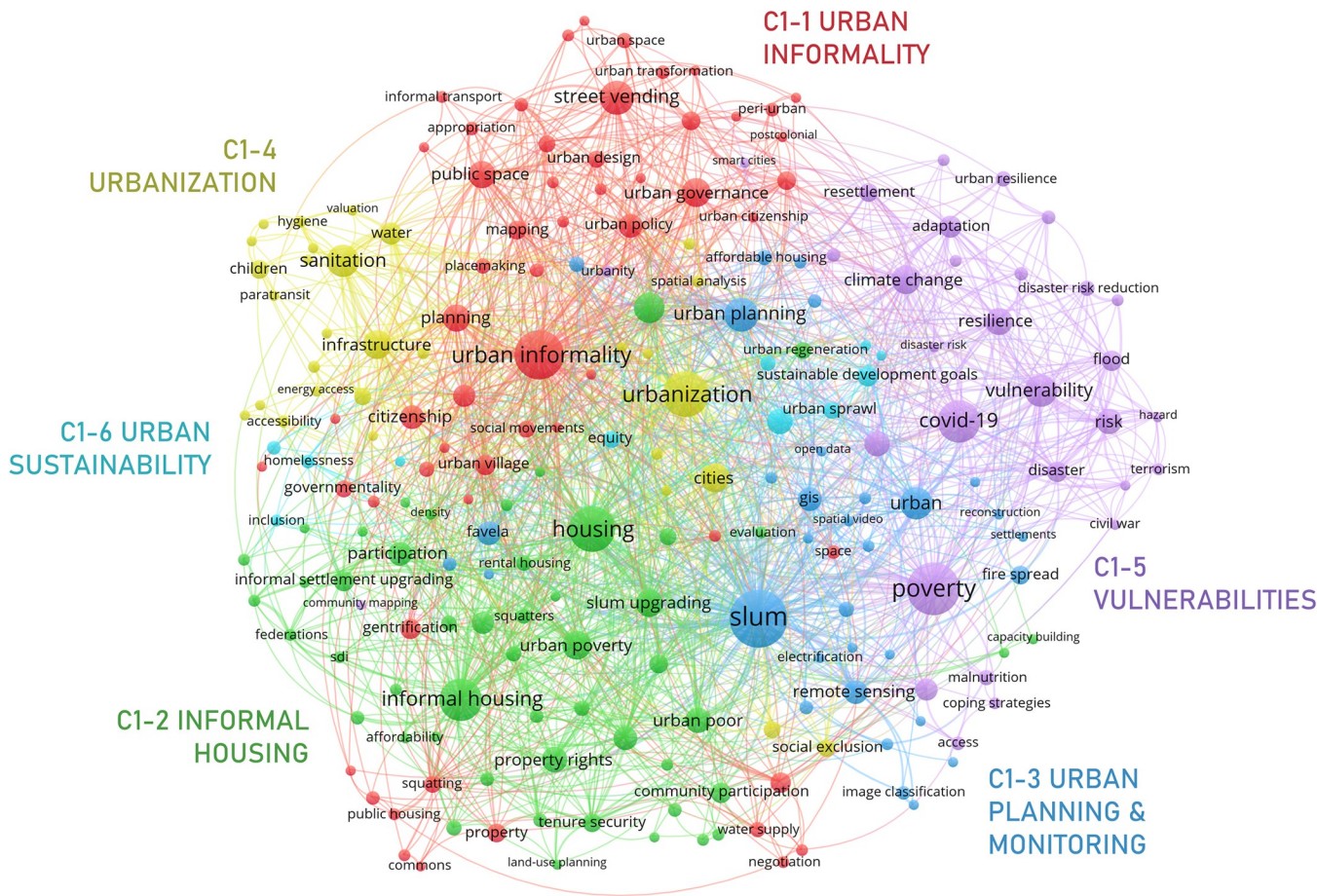

**Fig 7. Informal settlements network.**

strategies promoted by the government ('informal settlements upgrade', 'slum upgrading', 'urban redevelopment') and civil society ('participation' 'citizen-state relations') to improve housing conditions. Normative ('tenure security', 'land titling', 'property rights') and economic ('affordability', 'housing finance') requirements of adequate housing are also clearly relevant [57].

Cluster C1-3 groups nodes related to the use of technologies and methods ('gis', 'remote sensing', 'machine learning') for 'urban planning' and 'urban management' in 'informal settlements' [58–61]. These technologies and methods help tackle a variety of urban challenges, such as 'segregation', 'fire spread', 'water access', and 'electrification'.

C1-4 analyzes the effect of informal urbanization on a city's 'infrastructure'. Roy [62], for example, shows how the rapid growth of cities promotes informal urban practices, while Jones [63], on the contrary, points out the need to consider the growth of informal settlements and slums in the development of sustainable urbanization policies and strategies. Most literature in this micro cluster discusses the challenges ('accessibility', 'water quality', 'access to energy') related to different types of infrastructure ('sanitation', 'energy', 'transportation') in the context of informal urbanization (e.g., [64, 65]).

C1-5 discusses some of the main 'vulnerabilities', 'risk' and 'hazard' faced by residents of informal settlements and the factors that promote them: high population density, urbanization in hazard-prone areas, widespread 'poverty', precarious infrastructure, and limited access to public services. Rapid (unplanned) urbanization and climate change have increased the frequency and magnitude of natural disasters [66]. Thus, an important part of the literature analyzes community responses ('resilience', 'adaptive capacity') to these threats [67, 68] and more general adaptation strategies ('resettlement', 'climate change adaptation') [69]. The effects of poverty and other vulnerabilities on 'urban health' [70] and access to healthcare [71] have also been analyzed. Recently, the 'covid-19' pandemic has become a new challenge for informal settlements [72] that raises the need to rethink city planning [73] and urban governance [74].

The C1-6 micro-cluster focuses on the challenges posed by informal 'urban sprawl' to 'sustainable development' [75]. On the one hand, it group initiatives aimed at measuring and improving the conditions of informal settlements ('urban upgrading', 'sustainable development goals') [76, 77], particularly, in relation to the challenges and opportunities of the energy transition ('renewable energy') and climate change ('Paris agreement'); on the other hand, it discusses the role of the informal economy in the development of sustainable urbanism ('green economy', 'inclusion', 'equity') [78].

**3.3.2. Informal institutions.**   The "informal institutions" meso cluster, C2 (Fig 8), comprises nodes pertaining to interconnected aspects of non-standard or formally sanctioned dynamics of regulation and governance in multiple social settings. From an institutional point of view, informality is linked to institutional configurations and practices both with negative and positive social outcomes. Context, then, is important to determine how to value the effect of institutional informality. This meso cluster includes five micro clusters: C2-1 *informal market*, C2-2 *informal networks*, C2-3 *formal institutions*, C2-4 *governance*, and C2-5 *corruption*.

C2-1 and C5 thematically overlap (Fig 6). This micro cluster, however, depicts the institutional context of interaction in informal markets rather than their economic configuration and rationale. It shows a concern, first, with 'institutional voids' and other limitations of inadequate institutional transitions (e.g., 'capitalism', 'industrialization', 'modernization' [79, 80]), which fail to offer an adequate institutional context that allows for the replacement of informal types of 'trade' and 'exchange', such as 'barter' [81]. Because informal trade employs alternative institutional mechanisms of motivation and support (e.g., 'morality', 'religion', 'reputation' [82]), inadequate institutional transitions might foster 'conflict' and 'resistance', for example, around issues of 'land' tenure or 'commodification' [83], and might be more susceptible to the emergence of 'organized crime' and 'illegality' [84].

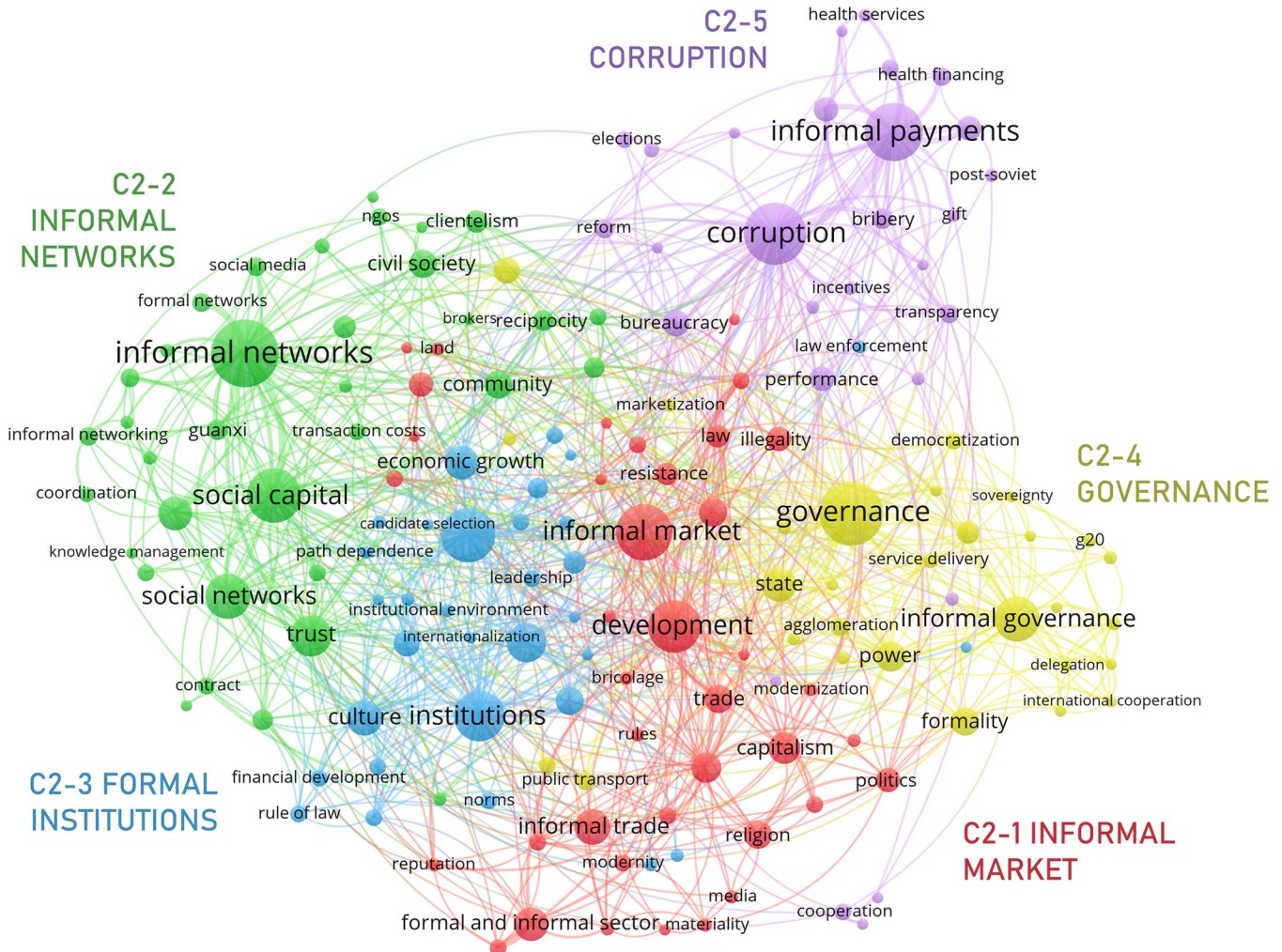

**Fig 8. Informal institutions network.**

C2-2 characterizes the institutional impact of informal 'social networks'. These networks are built through alternative institutional mechanisms e.g., 'trust' and 'reciprocity', in everyday 'informal practices'. They allow members of different 'communities', and the 'civil society', in general, to accumulate and mobilize 'social capital', which proves helpful for tasks such as 'knowledge management' [81, 85, 86]. Social networks can, as well, be used to achieve various social goals, such as increasing 'political participation' or enforcing 'social norms' [87, 88] (networks can also moderate violence and crime–see C6, below).

The C2-3 micro cluster centers on institutions themselves. This cluster is not thematically deep, for it includes several general nodes that strongly link out to other micro clusters: '[formal] institutions', 'formal and informal institutions'. It, however, shows the economic (e.g., 'economic growth', 'financial development', 'foreign direct investment'), political (e.g., 'leadership', 'corporate governance', 'decentralization') and regulatory ('rule of law', 'law enforcement', 'norms') advantages traditionally attributed to institutional formalization [89]. Interestingly, it also shows the primary role of 'culture' in the everyday operation of institutions and their (in)formality [90, 91].

C2-4 deals with different aspects of (informal) 'governance'. It explores, first, some targets of governance ('land reform', 'service provision', 'dispossession') that noticeably overlap with other dimensions of informality, especially the spatial [83, 92]. It also refers to the different levels of governance: local ('local government'), national ('state') and international ('brics', 'g20'). Lastly, this micro cluster evidences an interest in alternative governance mechanisms ('international cooperation', 'soft law' [93]) and in the institutional support for the exercise of (in)formal governance ('power', 'legality', 'sovereignty', 'legitimacy' [93, 94]]).

Finally, C2-5 depicts a two-fold connection between 'corruption' and informality, pertaining mostly to the effect of 'bureaucracy' on institutional 'performance'. It addresses, initially, the institutional role of 'informal payments' (with differing moral connotations e.g., 'gift', 'bribery', 'out-of-pocket payment'). These are common in contexts where individuals try to circumvent or take advantage of a limited institutional capacity to gain political favor or access social and public goods and services, especially 'health services' [95, 96]. Informality and corruption also intersect in the exercise of political power. Some types of corruption e.g., 'cronyism' or 'patronage', occur because individuals in a position of power take personal advantage of them, especially in regimes where 'accountability' and 'transparency' is low, leveraging informal channels and mechanisms to gain improper benefits and institutional advantages [97, 98].

**3.3.3. Informal employment enablers.** The meso cluster on "Informal employment enablers", C3 (Fig 9), is the first of three highly interconnected clusters predominately centered

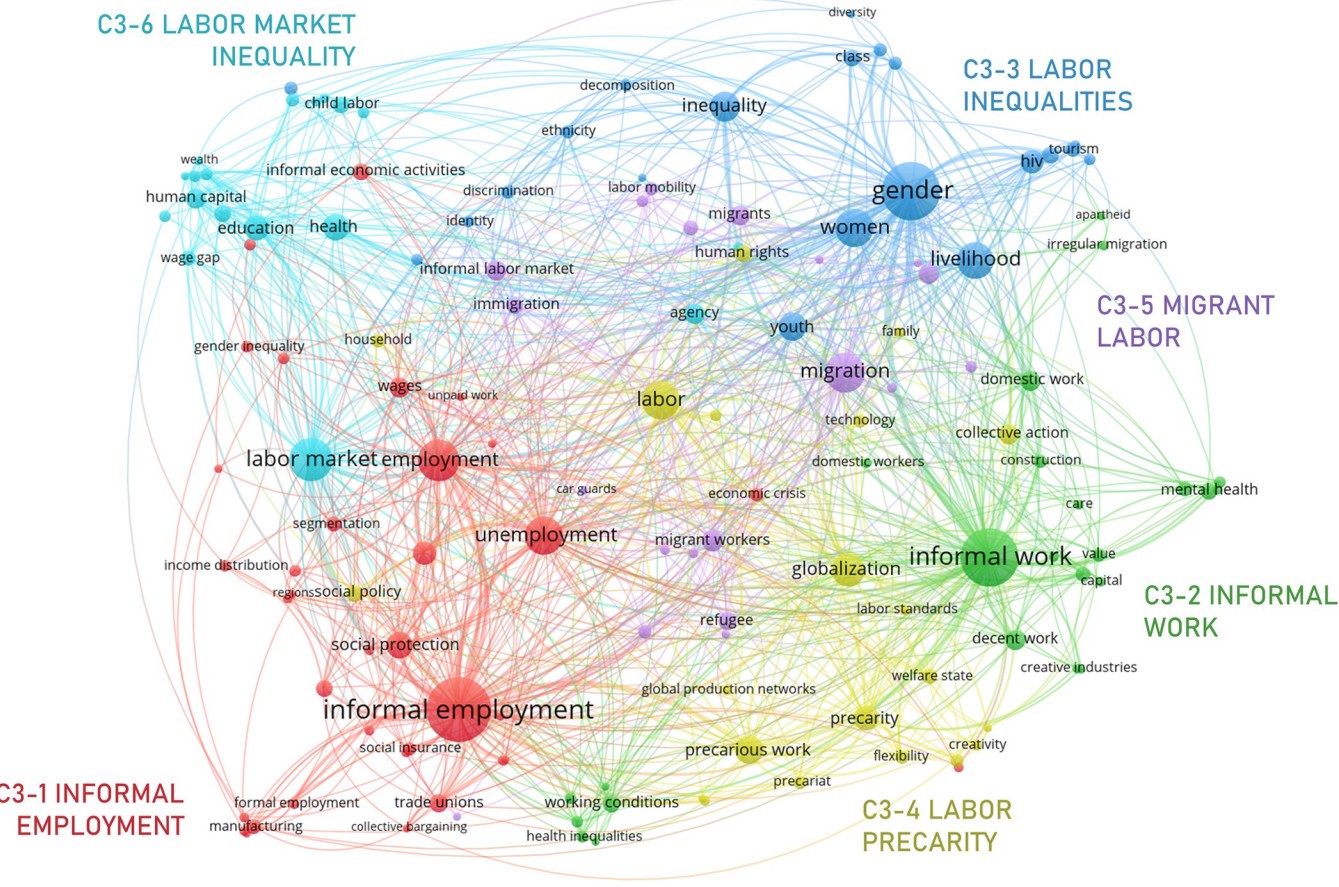

**Fig 9. Informal employment enablers network.**

on the economic dimension of informality (C4 and C5, described next, are the other two). C3 accounts for market informality from the perspective of the individual. Thus, its six micro clusters: C3-1 *informal employment*, C3-2 *informal work*, C3-3 *labor inequalities*, C3-4 *labor precarity*, C3-5 *migrant labor*, and C3-6 *labor market inequality*, group contextual traits that affect the individual experience of informality in diverse occupations and economic sectors.

Even though 'informal work' and 'informal employment' are often used interchangeably, the network structures of C3-1 and C3-2, the two largest micro clusters, suggest there is a thematic difference that might be worth considering. C3-1 pertains mostly to macroeconomic traits that lead or incentivize individuals to self-select into informality e.g., '[un]employment', '[minimum] wage', 'economic crisis', 'segmentation' [99], some typical features of informal employment (e.g., limited 'social protections/insurance') and its economic implications (e.g., 'income inequality/distribution') [100]. Alternatively, 'informal work' (C3-2) is most frequently used to address the 'working conditions' of an occupation or sector (e.g., 'domestic work', 'construction', 'gig economy'), and the effects of this type of work on informal workers, especially health-related ('mental health', 'well-being', 'depression') [101–103].

C3-3 groups at a finer level of granularity some interconnected individual traits that affect both self-selection into informality and the outputs of informal work. Interestingly, it also links these aspects to the 'livelihood' of informal workers [104]. This micro cluster includes aspects such as 'gender', 'class', 'ethnicity', age ('youth') or morbidity ('hiv'), which could become sources of labor inequality separately or in combination ('intersectionality', for instance, is part of this micro cluster) [105, 106].

Micro clusters C3-4 and C3-5 address two major dimensions of inequality in informal labor markets. The former groups nodes related to 'precarious work', which increases the incidence and prevalence of informality in economies and sectors where it was uncommon. It results from multiple interacting factors such as dynamics of 'globalization', the relaxation of 'labor standards', and the progressive dismantling of the 'welfare state' [107]. C3-5, complementarily, includes nodes pertaining to the connection between 'migration' and informality [108]. As migrants experience conditions of inequality and vulnerability, it is common for them to participate in the informal economy. Informal migrant work is often associated with distinctive dynamics of entry and '[labor]mobility', and economic outputs ('remittances') [109].

The last micro cluster: *labor market inequality* (C3-6), is not strongly connected thematically to the meso cluster. Its inclusion is due to the strong connection between the main nodes in this micro cluster and the main nodes in C3-1. Overall, it approaches sources of variation in the conditions of informal work through macro determinants of the 'labor market': 'income', 'health', 'education', 'human capital' [110]. It includes, however, some nodes that link more explicitly to contextual factors and determinants relevant for other micro clusters (e.g., 'child labor', 'youth employment', 'wage gap').

**3.3.4. Informal sector.** The "Informal sector" meso cluster, C4 (Fig 10), includes four micro clusters: C4-1 *informality in developing countries*, C4-2 *sustainable e-waste*, C4-3 *informal workers*, and C4-4 *occupational health*. It discusses the health risks that informal workers face due to the nature of their occupations, particularly in emerging and developing economies [111, 112], emphasizing on the lack of access to social protection given: (i) the diminished social security systems of these countries [113, 114] and (ii) the limitations that informal workers have to pay for these services [115].

Economic informality is pervasive in emerging and 'developing countr[ies]', in part, because of major formal-informal gaps in economic inputs and outputs [116]. A major cause is revealed by C4-1: these countries commonly experience problems with public service delivery [117, 118], which creates a need that is met by informal jobs like 'waste pickers'.

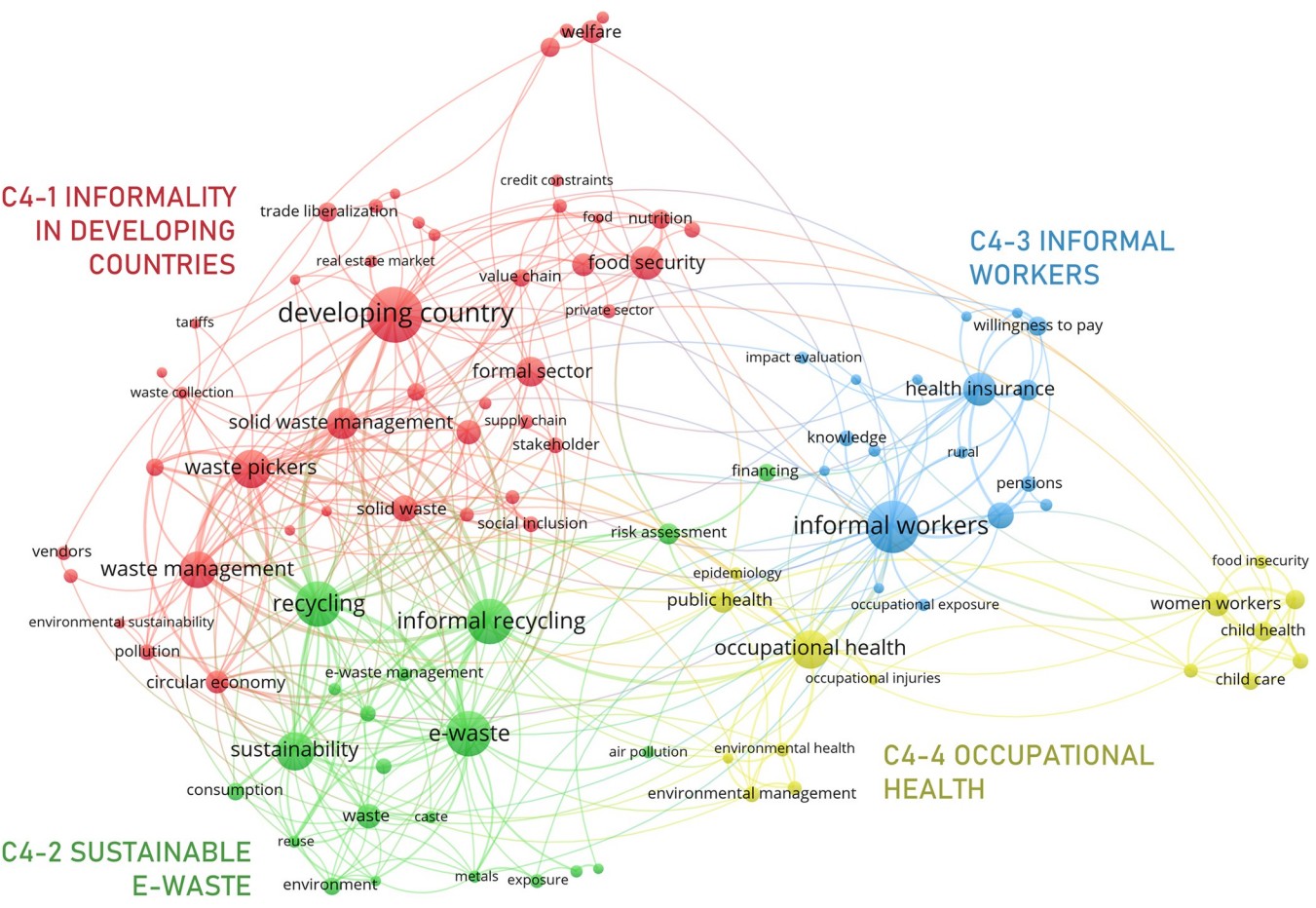

**Fig 10. Informal sector network.**

Furthermore, these nations frequently experience structural issues that are generally caused by poor regulation, unfavorable job circumstances and a lack of transparency regarding the procedures in economic activities with low productivity [118]. Some industries, such as 'agriculture' and 'real estate', among others, are prone to informality because of these structural problems.

C4-2 is closely connected with the nodes pertaining to 'waste management' in C4-1. It, however, focuses more on 'electronic waste' and the economic problems associated with it, making emphasis on the constraints for the formalization of such occupations [119]. It also addresses how e-waste is a major threat to 'sustainability' and how to cope with it at its source [112]. While e-waste has created a livelihood in many countries [111, 112], it poses some health risks for informal workers (e.g., due to 'exposure' to 'metals') and communities (e.g., due to the 'air pollution') [111, 120]. The disposal and recycling of 'e-waste' is a global 'environment[al]' challenge, with the greatest responsibility falling on the global production 'value chain' and the regulations that may be enacted by the countries that play a major role make in the global production network [121].

C4-3 discusses *informal workers*' health and social protection inequalities. Informal workers often lack 'health insurance' and other social protection benefits, such as 'pensions'. While COVID-19 has promoted several health reforms and has prompted some countries to provide health coverage to a population excluded from social security services [113, 114], the cluster

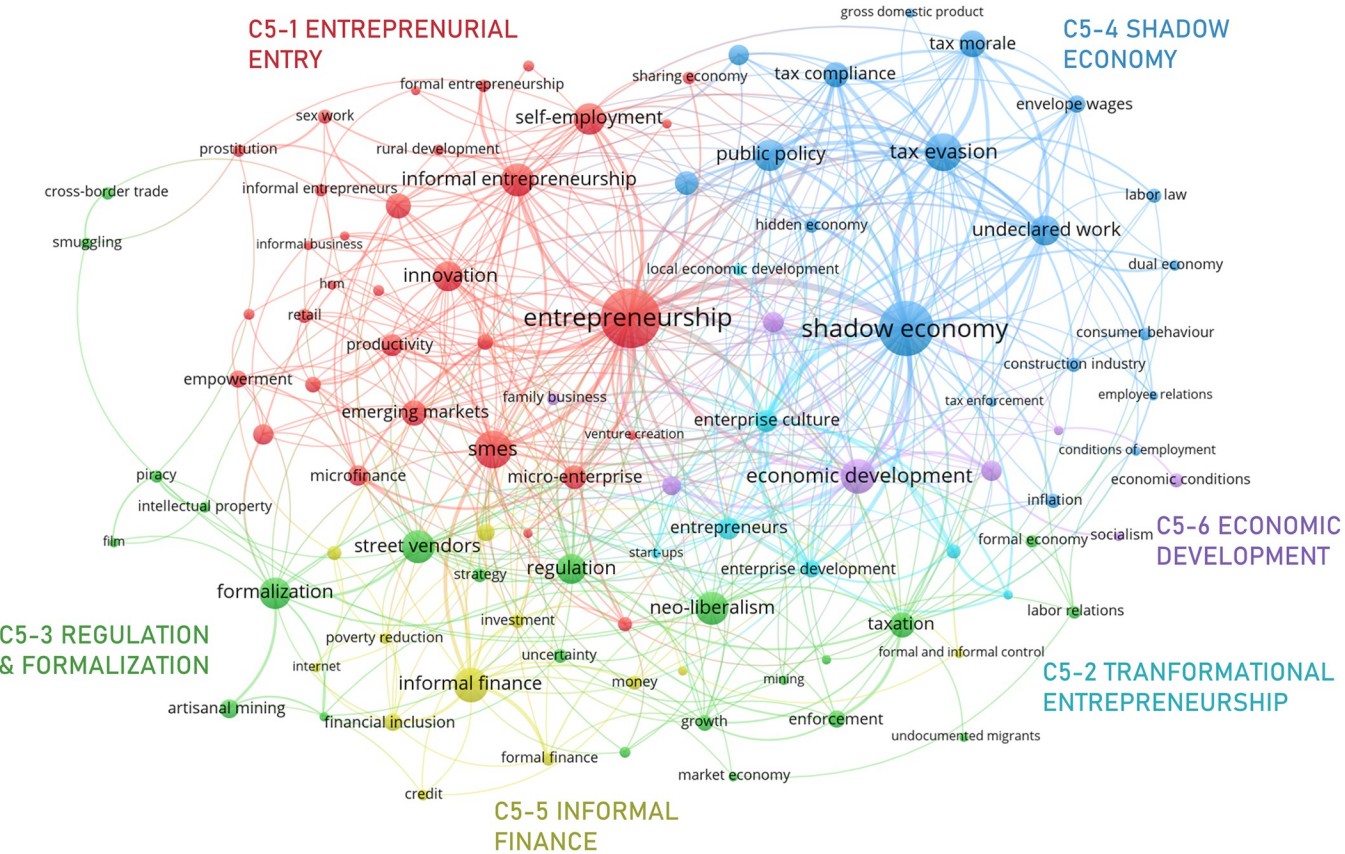

**Fig 11. Informal economy network.**

highlights the structural weaknesses of several countries in providing health care to their citizens and the challenges of implementing protective measures, especially for 'rural' and dispersed populations [113].

C4-4 provides an in-depth look at *'occupational health'* issues faced by informal workers and the environmental effect of these occupations. Building on C4-2, it shows that, overall, informal workers are more vulnerable to occupational risk and hazards [111, 120]. The lack of access to social health services makes informal workers more likely to become a 'public health'; problem in countries with high levels of informality [102]. These problems are intensified in the case of women in informal employment, as their children are also excluded from social security services, and they are not able to pay for 'childcare' while at work [115, 122].

**3.3.5. Informal economy.** As shown in Fig 11, C5: "informal economy" is comprised by six micro clusters: C5-1 *entrepreneurial entry*, C5-2 *transformational entrepreneurship*, C5-3 *regulation & formalization*, C5-4 *shadow economy*, C5-5 *informal finance*, and C5-6 *economic development*. This cluster includes topics that have shaped the traditional understanding in the economic sphere, such as 'tax morale', 'tax compliance', 'undeclared work', 'tax evasion', the 'shadow economy', among others. It includes, as well, a relatively newer interest in 'entrepreneurship'.

Entrepreneurship is accounted for from two perspectives: C5-1 centers on 'self-employment' and common barriers for *entrepreneurial entry*. Typical barriers are often classified into three dimensions: (i) the economic, comprising irregular income, undocumented finances, low productivity [123]; (ii) the political, encompassing activities prohibited by public

institutions, low or lack of regulation and supervision, 'tax evasion', ambiguity in legal status, non-existence of other formal associations [124, 125]; and (iii) the social, including survival motivations [126]. C5-2, alternatively, centers on entrepreneurship that has the potential for high value creation and formalization. Most entrepreneurships are initiated informally, as this practice allows entrepreneurs to explore the profitability of the sector without major costs. However, if medium and long-term profitability are high enough, informal entrepreneurs may choose to formalize, create value, foster innovation and promote employment [127].

The micro cluster C5-3 focuses on 'regulation', taxation models and 'formalization' strategies. Sepulveda et al. [128] indicate that informality is the result of a lack of knowledge about taxes and regulations, exclusionary systems, and cultural differences, such as ethnicity or migration status. Informal workers might not see major benefits in formalization, if it does not promise access to new markets or technologies [129]. Thus, strategies such as lowering the cost of business registration or relaxing regulations could only partially increase the likelihood of transitioning to formality [91, 130]. In some cases, more encompassing strategies addressing social dynamics such as 'migration' might be necessary [131].

C5-4 discusses the *shadow economy* and is centered on 'tax evasion', given the nature of organizations and their structure [132] and the magnitude and set-up of the tax burden [133]. Agents that avoid paying taxes usually have inefficient resources to benefit from lower capital costs and economies of scale [134]. Beyond the fact that tax evasion is a decision that agents make, there are certain conditions and factors that encourage this behavior, including political instability [132], poor public policy [135], moral hazard [2], regulations, prohibitions, and corruption.

C5-5 focuses on 'informal finance'. Low levels of 'financial inclusion' have historically encouraged individuals and business to enter and remain in the informal economy [136] and resort to various informal financial systems. Although most governments have tried to improve access to the formal financial systems, progress has been limited for a variety of reasons and, in several contexts, individuals, especially in vulnerable groups, remain dependent on informal financing [137]. In part, this is due to the fact that, even though informal financial systems tend to be less efficient and offer limited resources, they often better satisfy borrowing strategies of individuals that have been traditionally excluded from the formal financial system [138]. It is expected, however, that 'ICT' technologies will help give access to a wider range of formal financial services that might empower these individuals [139].

Finally, scholars usually discuss that 'economic development' is inversely proportional to informality [116, 117]. This means that the more formalization opportunities individuals and companies have, the lower the level of informality. This is especially true in emerging and developing economies. In this sense, C5-6 points out that this reduction in informality may require more focalized policies that target typical informal economic units in these economies, such as 'family businesses' [117].

**3.3.6. Informal social control.** The "social control" meso cluster, C6 (Fig 12), is the smallest and is only loosely integrated into the network (Fig 6). It addresses collective dynamics of control that are not exerted by the state's apparatus of control, but, rather, by agents (e.g., friends, coworkers, neighbors) and mechanisms (e.g., peer communication, rituals) that do not have ensuring norm compliance as their primary social role or function. This meso cluster groups four micro clusters: C6-1 *informal social control*, C6-2 *violence*, C6-3 *crime*, and C6-4 *control enforcement*.

C6-1 is the largest and most central micro cluster and revolves around the namesake node. It groups issues pertaining to organizational and functional aspects of 'informal social control'. The former is, first, addressed spatially, for structural and behavioral characteristics of spatial units, e.g., blocks, streets and, most frequently, 'neighborhood(s)' encourage the emergence of

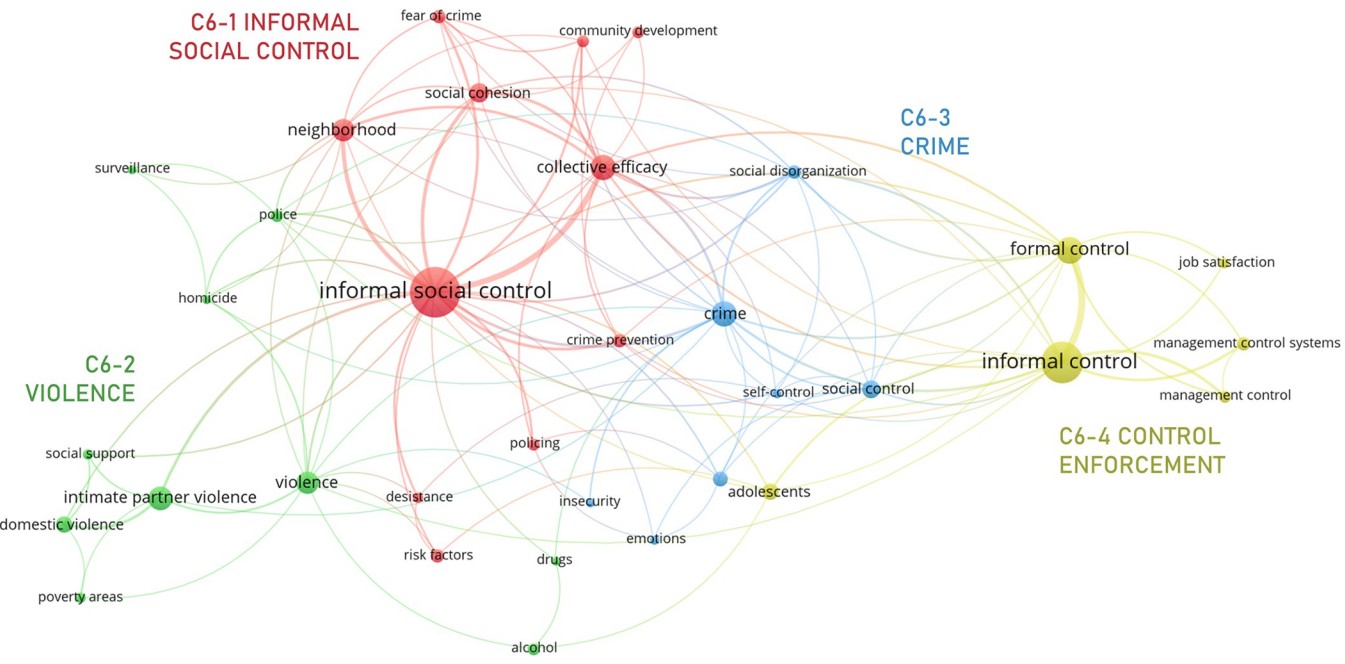

**Fig 12. Informal social control network.**

informal social control and influence its effectiveness [140, 141]. Organization is, as well, addressed socially, since the willingness to exert control i.e., the social group's 'collective efficacy', requires specific forms of social organization ('community development', 'social cohesion') [140, 142, 143]. Functional aspects of informal social control are mostly reflected by nodes depicting specific outcomes that are commonly targeted with these dynamics e.g., 'crime prevention' and 'desistance' [144, 145], as well as some of its motivations e.g., 'fear of crime' [146].

(C6-2) 'Violence' and (C6-3) 'crime' are pervasive concerns in contexts where the state has limited capacity for control. C6-2 includes nodes associated with typical individual ('drugs' and 'alcohol' [147, 148]) and social enablers ('poverty areas' [140]) of violence and victimization, potential deterrents (e.g., 'social support', 'police', 'surveillance' [141, 149]) and types of victimization ('homicide', 'intimate partner/domestic violence' [150, 151]). In recent literature, different forms of domestic violence are increasingly gaining relevance. While the household was not initially a spatial unit of interest, the social connections of the victim and the level of social organization of the community can help exert, in somewhat different manners, informal social control over domestic violence [151, 152]. C6-3, alternatively, accounts for how 'crime' is linked to general conditions of 'social disorganization' [153, 154] and '(in)security', individual and collective sources of control ('social control', 'self-control' [144, 155]), and some psychological factors involved in the regulation of crime e.g., 'emotions' [156].

Finally, C6-4 includes dynamics of informal control beyond the regulation of crime and victimization in urban areas, most prominently, in organizations and trading markets [157, 158]. A typical example of these dynamics of informal control is coworkers ostracizing deviant behavior at the workplace. The cluster also includes literature centered more generally on dynamics of 'management control' i.e., activities and processes, usually of vigilance and monitoring, that ensure the organization's goals are being adequately pursued [159], thus, the connection with 'job satisfaction' and 'management control systems'.

## 4. Discussion

Fig 13 introduces a concise depiction of the intellectual structure of the literature on socioeconomic informality that explicitly links higher- and lower-level clusters at the micro, meso and macro levels. It positions the 31 micro clusters within the six meso clusters that belong to the socioeconomic informality macro cluster. It offers, to our knowledge, the most comprehensive and synthesizing account of a corpus that is decidedly fragmented, because it receives inputs from a multiplicity of communities working separately. This taxonomy has the advantage that the resulting thematic categories emerge directly from the co-word analysis, instead of being defined a priori, as with most previous reviews (which tend to emphasize the economic and urban dimensions of informality).

Fig 14 presents the strategic diagrams for the six meso clusters. In the figure, the 31 micro clusters are classified according to their centrality and density. This adds to the understanding of the literature on socioeconomic informality by offering an account of the relative importance of each micro cluster in the current literature and their potential to (re)shape how socioeconomic informality is studied. The classification of the micro clusters according to their location in the strategic diagrams is presented in Table 3. Values for their size, connectedness, centrality and density are included.

Seven micro clusters are in QI (Motor themes) (Table 3). These themes have become mainstream, for they have achieved high levels of internal cohesion and are linked in several ways to other themes in the network. Their influence on the overall understanding of socioeconomic informality, however, is not justified for the same reasons. Two differences that are worth mentioning are scope and level. Regarding the former, the mainstream status of *vulnerabilities* (C1-5) appears to be driven by highly popular and transversal nodes, such as 'poverty' and 'covid-19', both strategically positioned at the meso-level border with C3 and C4. Even though it is part of C1, this micro cluster provides a thematically broad contribution to the understanding of socioeconomic informality. In contrast, the *informal networks* (C2-2) micro cluster has a more local orientation, being central to the institutional approach to informality. In terms of level, *shadow economy* (C5-4) and *crime* (C6-3), for instance, group thematically general nodes that are relevant for the macro-level characterization of the informal economy and informal social control, respectively. Comparatively, *labor inequalities* (C3-3) and *labor precarity* (C3-4) include nodes that mostly address micro-contextual determinants of socioeconomic informality.

Seven micro clusters are in QII (Ivory tower) (Table 3). The micro clusters in this quadrant have a high level of internal development, but do not have a significant thematic impact on the meso network. Many of these clusters share a common feature: they developed independently of the literature on informality and remain peripheral because they only partially overlap. *Occupational health* (C4-4) and *economic development* (C5-6) are topics with an overarching scope that are nonetheless important for informality. The former, for instance, is relevant because of the increased environmental health risks and hazards in some informal activities, such as waste picking (C4-2), and because of the limited access that informal workers have to health services (C4-3). An additional reason for the location of some of these micro clusters is that they may simultaneously account for alternative approaches to informality. The *governance* micro cluster (C2-4), for instance, addresses different dynamics of governance in the informal market, but accounts, as well, for informal or soft governance at the international level. Similarly, the *control enforcement* micro cluster (C6-4), as mentioned above, accounts for dynamics of informal (social) control in several social settings, which affect the criteria used to determine whether these forms of control are informal.

Eight micro clusters are in QIII (chaos/unstructured) (Table 3). Because of their low density and centrality, clusters in this quadrant are believed to be either emerging or declining. While

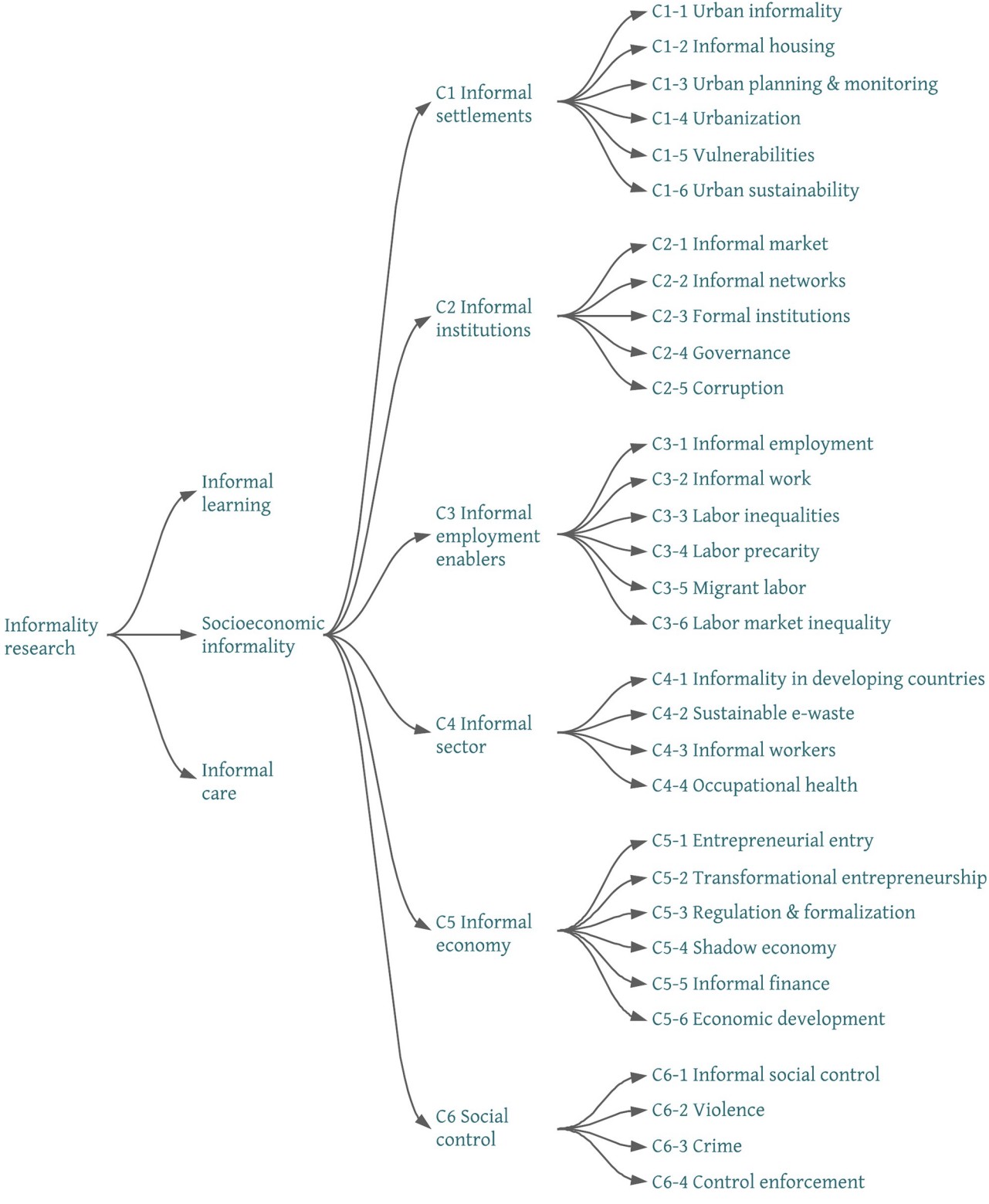

**Fig 13. A taxonomy of socioeconomic informality.**

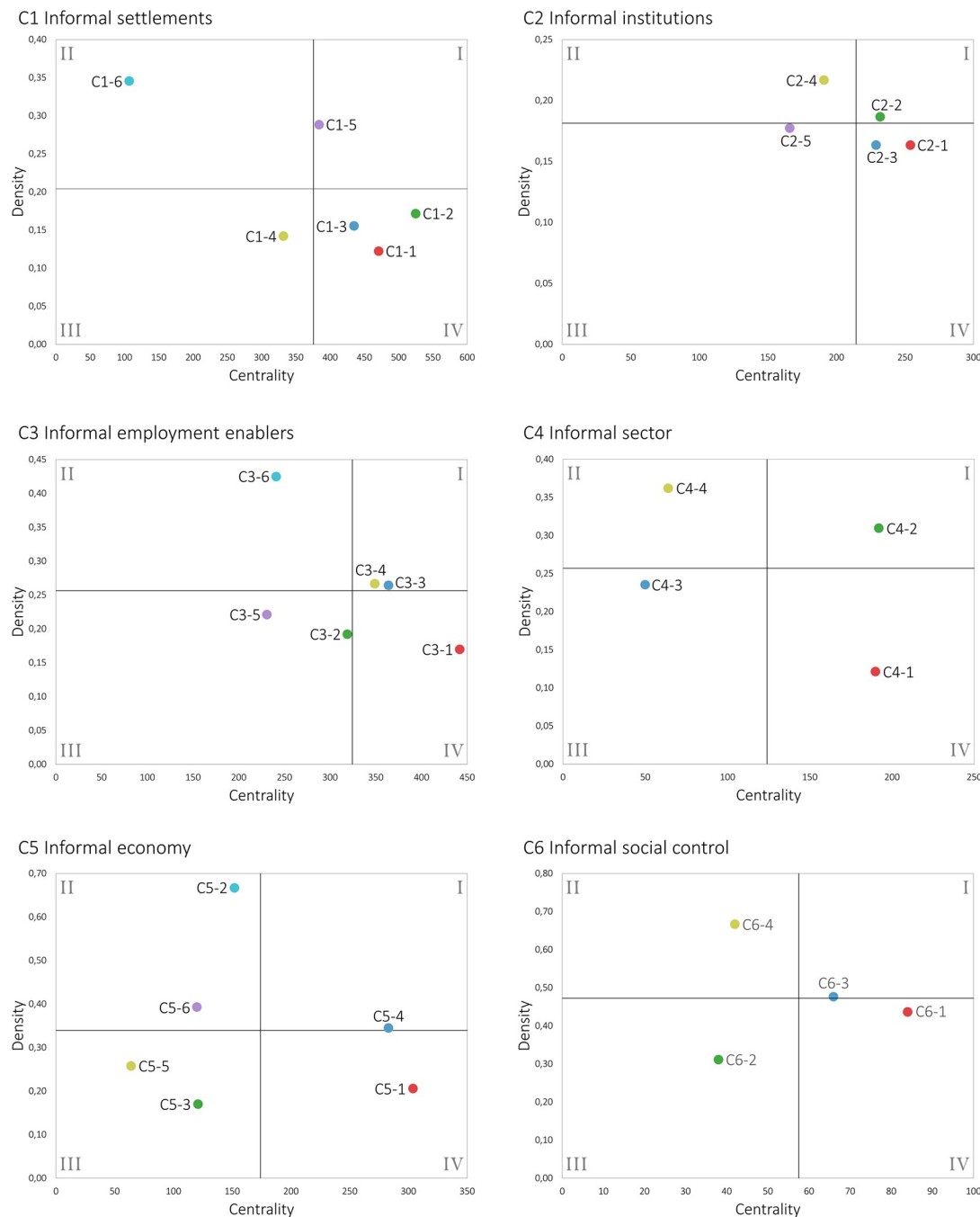

**Fig 14. Strategic diagrams for the 6 clusters of the meso level.**

the literature on socioeconomic informality is continuously increasing, two micro clusters could be declining: *urbanization* (C1-4) and *corruption* (C2-5). They are both dominated by relatively large and old nodes, 'corruption'/'informal payments' and 'urbanization'/'cities', respectively. Despite their relevance, the micro clusters are unstructured because, around these nodes, there is a constellation of smaller nodes with a narrower thematic focus and scope. In the *corruption* micro cluster, for example, the remaining nodes tackle specific types and

**Table 3. Strategic diagram metrics and micro cluster classification.**

| Q | Code | Micro cluster | No. nodes | No. edges | Centrality | Density |
|---|------|---------------|-----------|-----------|------------|---------|
| QI | C1-5 | Vulnerabilities | 31 | 134 | 384 | 0,2882 |
| | C2-2 | Informal networks | 35 | 111 | 232 | 0,1866 |
| | C3-3 | Labor inequalities | 22 | 61 | 364 | 0,2641 |
| | C3-4 | Labor precarity | 21 | 56 | 349 | 0,2667 |
| | C4-2 | Sustainable e-waste | 21 | 65 | 192 | 0,3095 |
| | C5-4 | Shadow economy | 19 | 59 | 283 | 0,3450 |
| | C6-3 | Crime | 7 | 10 | 66 | 0,4762 |
| QII | C1-6 | Urban sustainability | 11 | 19 | 107 | 0,3455 |
| | C2-4 | Governance | 29 | 88 | 191 | 0,2167 |
| | C3-6 | Labor market inequality | 18 | 65 | 241 | 0,4248 |
| | C4-4 | Occupational health | 15 | 38 | 64 | 0,3619 |
| | C5-6 | Economic development | 8 | 11 | 120 | 0,3929 |
| | C5-2 | Transformational entrepreneurship | 7 | 14 | 152 | 0,6667 |
| | C6-4 | Control enforcement | 6 | 10 | 42 | 0,6667 |
| QIII | C1-4 | Urbanization | 31 | 66 | 332 | 0,1419 |
| | C2-5 | Corruption | 28 | 67 | 166 | 0,1772 |
| | C3-2 | Informal work | 24 | 53 | 319 | 0,1920 |
| | C3-5 | Migrant labor | 20 | 42 | 231 | 0,2211 |
| | C4-3 | Informal workers | 18 | 36 | 50 | 0,2353 |
| | C5-3 | Regulation & formalization | 23 | 43 | 121 | 0,1700 |
| | C5-5 | Informal finance | 12 | 17 | 64 | 0,2576 |
| | C6-2 | Violence | 10 | 14 | 38 | 0,3111 |
| QIV | C1-1 | Urban informality | 58 | 202 | 471 | 0,1222 |
| | C1-2 | Informal housing | 50 | 210 | 525 | 0,1714 |
| | C1-3 | Urban planning & monitoring | 32 | 77 | 435 | 0,1552 |
| | C2-1 | Informal market | 39 | 121 | 254 | 0,1633 |
| | C2-3 | Formal institutions | 32 | 81 | 229 | 0,1633 |
| | C3-1 | Informal employment | 35 | 101 | 442 | 0,1697 |
| | C4-1 | Informality in developing countries | 45 | 120 | 190 | 0,1212 |
| | C5-1 | Entrepreneurial entry | 32 | 102 | 304 | 0,2056 |
| | C6-1 | Informal social control | 11 | 24 | 84 | 0,4364 |

contexts of corruption and informal payments. A narrower focus and scope is, at the same time, the reason the others micro clusters in the quadrant are better classified as emerging. They are, initially, highly contextual, highlighting the multiple dimensions of informality and how these dynamics hinge upon specific combinations of social, cultural, and economic variables. More important, though, they bring agency to the foreground: how individuals consciously and deliberately accumulate and mobilize individual and social resources to navigate their conditions of informality.

Nine micro clusters are in QIV (Bandwagon) (Table 3). This is the most populated quadrant, and, perhaps, the one with the most interesting dynamics. It groups clusters that are transversal to the literature, but not entirely cohesive internally. Similarly to declining themes in QIII, the location of some micro clusters in QIV e.g., *informal employment* (C3-1) and *informality in developing countries* (C4-1), may be determined by the behavior of high-occurrence nodes. What makes these clusters transversal, instead of declining, is that these high-occurrence nodes remain current or are more strongly connected to a diversity of other nodes. The location of other micro clusters e.g., *formal institutions* (C2-3) and *entrepreneurial entry* (C5-1), is, instead, more likely attributable

to the existence of second-tier nodes that are comparatively newer or with higher frequency, and also link out more significantly to other micro clusters in the meso network.

Overall, the strategic diagrams offer some insights into the thematic relevance and influence of the 31 micro clusters and, to a certain extent, provide a rationale for the current state of the meso network. While the current location of the micro clusters can be attributed to a few common reasons, a structural change in the meso network should be expected in the future given that micro clusters located in the same quadrant for, arguably, the same reason, may behave differently in the future. For example, *entrepreneurial entry* (C5-1) could easily gain centrality and move to QI due to the increasing interest in informal entrepreneurship, its overall contribution to economic development and its potential for growth and innovation. Conversely, *formal institutions* (C2-3) is more likely to remain in QIV, or even move to QIII, since its central nodes focus on very general and standard institutional dynamics.

The future structure of the meso network is also likely to change due to some thematic concerns that are downplayed by its current node composition. Sustainability, for instance, is progressively becoming important for every dimension of socioeconomic informality. In the current network structure, however, this contemporary relevance is not as evident, for it is split among seven nodes in different micro clusters. Since some aspects of sustainability are transversal to the multiple contexts of socioeconomic informality, it is possible that a structural unification occurs in the future, driven, initially, by the 'sustainability' node, which is currently positioned between C4 and C1.

A more general and interesting dynamic with the potential to reshape the meso network is the increasing awareness about, and thematic accommodation of, the effects and relevance of micro and local aspects of socioeconomic informality. Several of the most novel nodes, some of the micro clusters in QIII and, arguably, the entire C3 meso cluster bring to the foreground contextual traits of actors, occupations and dynamics traditionally considered informal. As the acknowledgement that informality has a distinctive thematic identity beyond its relationship (or contrasts) with formality becomes widespread, a change in the thematic structure of the corpus should be expected. This change may be driven, first, by an increasing research output emphasizing the diverse contextual traits of informality, second, by the potential development of more specialized terminology (starting with the popularization of additional terms with the prefix "informal") and, third, by the conceptual clarification of the several socioeconomic informalities (e.g., those pertaining to governance).

Finally, future changes in the meso network could also be anticipated by paying attention to some nodes that, while not entirely general (e.g., 'poverty'), seem to be able to thematically link or branch out to several micro and meso clusters. 'Street vending', for example, connects the economic aspects of the occupation (spread among the three clusters with economic underpinnings) with the spatial and institutional aspects linked to its dependence on the urban public space. These nodes or constellations of nodes might be used not only to anticipate the future state of the corpus, but also to identify areas of common work between the different communities interested in socioeconomic informality.

## 5. Conclusion

Despite its high incidence and prevalence, especially in emerging and developing countries, the current understanding of informality is severely fragmented, in part, due to the variety of phenomena to which this label applies and the multiple disciplinary communities that study them. Although various authors have attempted to develop taxonomies and classifications that better grasp informality's scope and nature, these efforts hinge on predefined frameworks and criteria that render only a partial characterization of socioeconomic informality. In this study,

we avoided any frame of reference in the delimitation of what counts as socioeconomic informality, letting it, instead, to emerge from a co-word analysis of the literature available in Scopus from the 1960s until 2021.

The analysis showed that, at the macro level, there are three sufficiently separated themes that respond to alternative conceptualizations and understandings of informality and the formal-informal dichotomy: (i) informal education, (ii) informal care, and (iii) socioeconomic informality. In this third macro cluster, six meso-themes were identified: (1) informal settlements, (2) informal sector, (3) informal economy, (4) informal institutions, (5) informal employment enablers, and (6) social control.

The six meso clusters, in turn, were decomposed into 31 micro clusters that provide the most comprehensive characterization of informality in the socioeconomic dimension. In general, informality is about individuals, communities and institutions using *alternative* means and mechanisms to experience and navigate a variety of social settings. The analysis, however, shows that these *alternative* means and mechanisms are not homogeneous, nor share an identity. What makes them 'informal', the micro clusters suggest, is, rather, that they are not 'formal'—and not being formal manifests in multiple ways e.g., being non-official, non-standard, non-traditional, or non-sanctioned.

Since there is more than one way in which phenomena can be informal, socioeconomic informality can have positive and negative effects. The node composition of the micro clusters supports previous findings that link informality to higher vulnerabilities and inequalities. For instance, issues of infrastructure, exposure to natural disasters and poverty, among others, are prominent in the six micro clusters in the "informal settlements" meso cluster. Similarly, problems with social protection and working arrangements and conditions are noticeable in the three meso clusters addressing economic informality. Multiple positive effects of informality are equally identifiable, though. Informal finance, for example, partially counters the negative effects of financial exclusion. In turn, informal social networks facilitate the accumulation and mobilization of various sorts of capital. Finally, informal social control deters crime and victimization in contexts where the state's capacity is limited.

A recent trend in the literature that could help provide a more nuanced understanding of these various *informalities* is the increasing interest in the contextual determinants and effects of informality. This more contextual orientation in the analysis of informality, visible both in the node composition of the micro and meso clusters and the strategic diagram, might prove useful, first, because, in practice, formal and informal are not neatly separated and, second, because, for most phenomena of interest, 'informal' is too wide of an umbrella concept.

Since most individuals experience informality in more than one way, accounting more explicitly for context should also offer means to bridge the literatures and communities working on socioeconomic informality. It would not be hard to find individuals, for example, that work informally, live in informal settlements, use informal transport, rely on informal finance, and make informal payments to access some services. Although connections between the different literatures could be established in a variety of ways, large nodes at the intersection between meso clusters (e.g., 'gender', 'employment', 'education', 'migration', 'poverty', 'corruption', and 'governance'), as well as themes that are currently fragmented in multiple nodes (e.g., sustainability) could pave the way for integration.

## Supporting information

**S1 File. Records extracted from Scopus for the search statement (I).** This file lists the 29546 documents used for the macro analysis.
(XLSX)

**S2 File. Records extracted from Scopus for the search statement (II).** This file lists the 29546 documents used for the meso and micro analyzes.
(XLSX)

## Author Contributions

**Conceptualization:** Nelson Alfonso Gómez-Cruz, David Anzola, Aglaya Batz Liñeiro.

**Data curation:** Nelson Alfonso Gómez-Cruz, David Anzola, Aglaya Batz Liñeiro.

**Formal analysis:** Nelson Alfonso Gómez-Cruz.

**Funding acquisition:** Aglaya Batz Liñeiro.

**Methodology:** Nelson Alfonso Gómez-Cruz.

**Project administration:** Aglaya Batz Liñeiro.

**Supervision:** Nelson Alfonso Gómez-Cruz, Aglaya Batz Liñeiro.

**Validation:** Nelson Alfonso Gómez-Cruz, David Anzola, Aglaya Batz Liñeiro.

**Visualization:** Nelson Alfonso Gómez-Cruz.

**Writing – original draft:** Nelson Alfonso Gómez-Cruz, David Anzola, Aglaya Batz Liñeiro.

**Writing – review & editing:** Nelson Alfonso Gómez-Cruz, David Anzola, Aglaya Batz Liñeiro.

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
