## [Decision Letter · Decision Letter 0]

16 May 2023

PONE-D-23-06615Unveiling the intellectual structure of informality: Insights from the socioeconomic literaturePLOS ONE

Dear Dr. Gómez Cruz,

Thank you for submitting your manuscript to PLOS ONE. After careful consideration, we feel that it has merit but does not fully meet PLOS ONE’s publication criteria as it currently stands. Therefore, we invite you to submit a revised version of the manuscript that addresses the points raised during the review process.

We look forward to receiving your revised manuscript.

Kind regards,

Yue Gong

Academic Editor

PLOS ONE

Journal Requirements:

"This research has been developed in the framework of the Colombia Científica – “Alianza EFI” Research Program, with code 60185 and contract number FP44842-220-2018, funded by The World Bank through the call Scientific Ecosystems, and managed by the Colombian Ministry of Science, Technology and Innovation."

"Grant number: FP44842-220-2018

Founder: Colombia Científica – “Alianza EFI”

URL: https://alianzaefi.com/

Reviewers' comments:

Reviewer's Responses to Questions

**Comments to the Author**

1. Is the manuscript technically sound, and do the data support the conclusions?

Reviewer #1: Yes

2. Has the statistical analysis been performed appropriately and rigorously? 

Reviewer #1: Yes

3. Have the authors made all data underlying the findings in their manuscript fully available?

Reviewer #1: Yes

4. Is the manuscript presented in an intelligible fashion and written in standard English?

Reviewer #1: Yes

5. Review Comments to the Author

Reviewer #1: The article approaches an interesting issue.

The abstract is not very well developed since there are not clearly presented some ideas regarding the results as well as the main conclusion. Also, the chosen methodology could have been better, briefly, descripted.

Introduction is under-developed. The state-of-art, the research questions etc. should be better explained/ described and sustained by relevant and actual references.

The methodology is well described. There are however some shortcomings that must be corrects. By example, in section 2.1 - equations 1 and 2 are not "equations". Please select another term for them.

Results section is over-extended. Some figures are not clear (Figure 3 is not clear. Please improve the figure quality.

Idem figure 6.). Parts of this section could be transferred to Discussions section.

The axis of figure 14 are also not clear.

Overall, the article is a good one which, with some minor corrections, can be accepted for being published.

6. PLOS authors have the option to publish the peer review history of their article (what does this mean?). If published, this will include your full peer review and any attached files.

Reviewer #1: No

---

## [Author Response · Author response to Decision Letter 0]

16 Aug 2023

Dear reviewer

I hope you are well. We sincerely thank you for your time and dedication in reviewing our article entitled " Unveiling the intellectual structure of informality: Insights from the socioeconomic literature" that was submitted to PLoSONE journal. We appreciate your comments and suggestions, which have been of great help in improving the quality of our work.

We have made the suggested modifications in the manuscript in accordance with their recommendations. Below, we describe how we have addressed each of the points shown:

Reviewer comment: “The abstract is not very well developed since there are not clearly presented some ideas regarding the results as well as the main conclusion. Also, the chosen methodology could have been better, briefly, descripted”.

Response: The abstract was expanded to more precisely describe the methodology, specify the results and present the main conclusion. Below are the highlighted modifications:

In the socioeconomic sphere, the concept of informality has been used to address issues pertaining to economic dynamics, institutions, work, poverty, settlements, the use of space, development, and sustainability, among others. This thematic range has given way to multiple discourses, definitions and approaches that mostly remain within a single phenomenon and conform to traditional disciplinary lines, making it difficult to fully understand informality and adequately inform policymaking. In this article, we carried out a multilevel co-word analysis with the purpose of unveiling the intellectual structure of socioeconomic informality. Co-occurring document keywords were used, initially, to delimit the scope of the socioeconomic dimension of informality (macro level) and, later, to identify its main concepts, themes (meso level) and sub-themes (micro level). Our results show that there is a corpus of research on socioeconomic informality that is sufficiently differentiable from other types of informality. This corpus, at the same time, can be divided into six major themes and 31 sub-themes related, more prominently, to the informal economy, informal settlements and informal institutions. Looking forward, the analysis suggests, an increasing focus on context and on the experience of multiple ‘informalities’ has the potential, on the one hand, to reveal links and unify this historically fragmented corpus and, on the other hand, to give informality a meaning and identity that go beyond the traditional formal-informal separation.

Reviewer comment: “Introduction is under-developed. The state-of-art, the research questions etc. should be better explained/ described and sustained by relevant and actual references.”

Response: The introduction was extended to further develop the state of the art and new and recent references were incorporated. Likewise, the research question was integrated. Below are the highlighted modifications: 

Socioeconomic informality is a complex, multidimensional and context-dependent phenomenon in which social, economic, political, and geographical variables intervene (Polese, 2023). The composition and impact of informality vary across economies and regions (DellAnno, 2021). In emerging and developing countries, informal work accounts for around 30-35% of GDP and 70% of the total employment (Loayza, 2016). Globally, informal employment reaches 61.2% (ILO, 2018), which represents close to 2 billion people. Similarly, it is estimated that one in eight people in the world lives in informal settlements under precarious conditions (ONU-Habitat, 2016). In urban areas, informal settlements shelter 25% of the population (ONU-Habitat, 2013). Consequently, informality has profound implications for global development and sustainability.

Research on informality has been characterized by its compartmentalized development, with a predominant focus on specific themes such as informal settlements, the informal economy, urban informality, the informal labor market, informal employment, informal institutions, and others (Polese, 2023; Boanada-Fuchs & Boanada, 2018). Unfortunately, this fragmented approach has resulted in a lack of interconnectedness between these themes, obscuring their potential linkages at the core of various socioeconomic issues. As a result, several authors have attempted to develop taxonomies or classifications of informality, aiming to improve the understanding of the primary factors contributing to it. However, these efforts have arisen from diverse thematic perspectives.

For example, it is common to find general analyzes on the informal economy (Dell’Anno, 2021; Polese, 2023) or on informal settlements (Wekesa et al, 2011). Informality, however, has consistently been approached from a singular, narrowly framed perspective. Boanada-Fuchs & Boanada-Fuchs (2018) concentrated their study on publications related to ‘informal economy’, ‘housing’, ‘land tenure’ and ‘urban planning’, while Venerandi & Mottelson (2021) proposed a taxonomy exclusively focused on categorizing ‘informal settlements‘. Furthermore, Dovey et al. (2020) acknowledge that certain patterns lead to the formation of informal settlements and influence the emergent urban design or informal morphogenesis. Alsayyad (2004) discusses various patterns of urban informality, such as invasion, survival, assimilation, adaptation, and cooperation. Roy (2005) delves into the epistemology of public policies employed in urban planning and presents a characterization of these policies. Although most approaches have traditionally centered on the topic of informal settlements, Fernandez & Villar (2016) put forward a taxonomy to better understand the informal labor market by analyzing three characteristics: choice, productivity, and barriers. However, efforts to establish general frameworks are scarce and are often focused on ad hoc definitions or on dimensions that fall within the domain of expertise of the authors (Boanada-Fuchs & Boanada, 2018; Polese, 2021).

This lack of thematic unification makes it difficult to (i) clarify informality’s borders, dimensions, and scope, (ii) achieve a basic conceptual understanding, (iii) offer satisfactory measurement alternatives and, in general, (iv) sufficiently integrate the research areas for which the concept is important. Hence, this paper endeavors to address the question: what is the intellectual structure of informality research?

To address this question, we employed a multilevel variant of co-word analysis (Callon et al., 1986), which enables us to minimize bias in the development and classification of topics addressed in the literature. By using this approach, we aim to reveal the central research problems related to socioeconomic informality. Co-word analysis is a quantitative content analysis technique (Zhang et al., 2017; He, 1999) that allows mapping the intellectual structure of a research field through the relationships between concepts (co-occurrences) within a corpus of literature. In this text, we use the method at three different levels of detail to (i) determine the frontiers of research on socioeconomic informality and its differences with other forms of informality (macro level), (ii) identify the main concepts, themes (meso level) and sub-themes (micro level) addressed in informality studies, (iii) explore the relationships between the various sub-themes and (iv) assess, from a general perspective, the relative importance and level of development of the various sub-themes. 

To achieve this goal, the paper is organized as follows. Section 2 provides an overview of the methodological framework. Section 3 presents the main results of the multilevel co-word analysis. Initially, it establishes the boundaries of socioeconomic informality (macro level). Subsequently, its main themes and sub-themes are identified and characterized (meso and micro levels). Section 4 introduces a taxonomy that succinctly captures the intellectual structure of socioeconomic informality and discusses its evolution and future opportunities. Finally, a conclusion is provided in section 5.

Reviewer comment: “The methodology is well described. There are however some shortcomings that must be corrects. By example, in section 2.1 - equations 1 and 2 are not "equations". Please select another term for them.”

Response: The expression “search equation” has been consistently replaced throughout the text by “search statement”. Additionally, a thorough review of the methodological apparatus was carried out. Some additional shortcomings were detected and fixed.

Reviewer comment: “Results section is over-extended […]. Parts of this section could be transferred to Discussions section.”

Response: The results section was fully reviewed. However, its structure and contents are eminently descriptive. For this reason, we made an effort to reduce the length of the section, rather than transfer contents to the discussion, since they do not apply there.

Reviewer comment: “Figure 3 is not clear. Please improve the figure quality. Idem figure 6. […] The axis of figure 14 are also not clear.”

Response: All the figures were reconstructed following the guidelines of the journal in terms of size and resolution. The size of the fonts has also been revised. New versions are attached.

We have worked meticulously on the requested modifications, ensuring that the manuscript is clearer, more rigorous and valuable to the scientific community. Following the guidelines of the journal, we attach a revised version of the manuscript that incorporates all suggested modifications. In addition, we attach another version in which the revised parts of the document are highlighted to facilitate their identification. We hope these revisions meet your expectations and that the article is now suitable for publication in PLoSONE.

We thank you again for your time and effort in reviewing our work. We value your comments and opinions. We hope that our response and the modifications made are satisfactory. We remain at your disposal for any additional comments or questions you may have.

---

## [Decision Letter · Decision Letter 1]

2 Nov 2023

PONE-D-23-06615R1Unveiling the intellectual structure of informality: Insights from the socioeconomic literaturePLOS ONE

Dear Dr. Gómez Cruz,

Thank you for submitting your manuscript to PLOS ONE. After careful consideration, we feel that it has merit but does not fully meet PLOS ONE’s publication criteria as it currently stands. Therefore, we invite you to submit a revised version of the manuscript that addresses the points raised during the review process.

The additonal two reviewers provided posivitve comments. Please revise the methodology section according to the reviewer 2’s comments. Please ensure that your decision is justified on PLOS ONE’s publication criteria and not, for example, on novelty or perceived impact.

We look forward to receiving your revised manuscript.

Kind regards,

Yue Gong

Academic Editor

PLOS ONE

Journal Requirements:

Reviewers' comments:

Reviewer's Responses to Questions

**Comments to the Author**

1. If the authors have adequately addressed your comments raised in a previous round of review and you feel that this manuscript is now acceptable for publication, you may indicate that here to bypass the “Comments to the Author” section, enter your conflict of interest statement in the “Confidential to Editor” section, and submit your "Accept" recommendation.

Reviewer #2: (No Response)

Reviewer #3: All comments have been addressed

2. Is the manuscript technically sound, and do the data support the conclusions?

Reviewer #2: Yes

Reviewer #3: Yes

3. Has the statistical analysis been performed appropriately and rigorously? 

Reviewer #2: N/A

Reviewer #3: Yes

4. Have the authors made all data underlying the findings in their manuscript fully available?

Reviewer #2: Yes

Reviewer #3: Yes

5. Is the manuscript presented in an intelligible fashion and written in standard English?

Reviewer #2: Yes

Reviewer #3: Yes

6. Review Comments to the Author

Reviewer #2: It is my pleasure to provide my insights on the article titled "Unveiling the Intellectual Structure of Informality: Insights from the Socioeconomic Literature."

Having thoroughly examined the content, it is evident that the article has undergone a revision process. Building upon the authors' efforts, I aim to offer constructive feedback, with the intention of enhancing the scholarly robustness of the work.

Beginning with the Introduction, while succinct, it effectively addresses the critical elements by delineating the research gap and posing the central research question. The section concludes by outlining the subsequent chapters, encompassing the pivotal aspects of the discourse.

In terms of the methodology, it is imperative for the authors to explicate the rationale behind the selection of the Elsevier Scopus scientific database. The mere mention of its status as the largest curated database worldwide remains insufficient. It prompts the question of the exclusion of the Web of Science. It would be beneficial to explore literature comparing the strengths and weaknesses of both databases, subsequently providing a well-founded justification for the utilization of Scopus. Furthermore, the decision to solely rely on a single database warrants clarification, particularly regarding the transparency and replicability of the research. Elucidating how this approach facilitates ease of study replication by other researchers is imperative. Additionally, an explanation for not utilizing search engines such as Google Scholar is necessary, possibly highlighting the challenges in applying filters due to the inclusion of non-peer-reviewed literature. Moreover, the incorporation of a research protocol, such as PRISMA, could have potentially strengthened the methodology by facilitating effective data filtration.

Moving on to the results and discussion, a thorough and coherent presentation significantly contributes to the scientific merit of the work. However, the conclusion could be enhanced by restructuring it into subsections, such as contributions to existing theoretical knowledge, managerial implications, research limitations, and prospects for future research. While this proposed restructuring is not obligatory, it would undoubtedly augment the article's readability and facilitate a more comprehensive grasp of the implications drawn from the findings.

In summary, the article exhibits a commendable scholarly effort. However, before final publication, I strongly recommend bolstering the methodology section with a more comprehensive explication of the database selection and an exploration of the justifications underpinning this decision. The remaining recommendations primarily aim to reinforce the existing work, with an emphasis on minor adjustments and refinements.

In conclusion, I extend my congratulations to the authors and wish them success in their academic pursuits.

Reviewer #3: The authors have addressed and responded to all comments and suggestions given in the previous round of review. The reviewer has no further comments.

7. PLOS authors have the option to publish the peer review history of their article (what does this mean?). If published, this will include your full peer review and any attached files.

Reviewer #2: **Yes: **João Carlos Gonçalves dos reis

Reviewer #3: No

---

## [Author Response · Author response to Decision Letter 1]

17 Dec 2023

Dear editor and reviewers:

We have provided a detailed response to the suggestions provided by the reviewers in the "Response to reviewers" document.

Thank you very much for the review and for considering our research for publication in PLOSONE.

Kind regards,

The authors

---

## [Editor Report · Decision Letter 2]

9 Jan 2024

Unveiling the intellectual structure of informality: Insights from the socioeconomic literature

PONE-D-23-06615R2

Dear Dr. Gómez Cruz,

We’re pleased to inform you that your manuscript has been judged scientifically suitable for publication and will be formally accepted for publication once it meets all outstanding technical requirements.

Kind regards,

Yue Gong

Academic Editor

PLOS ONE

---

## [Editor Report · Acceptance letter]

22 Jan 2024

PONE-D-23-06615R2 

PLOS ONE

Dear Dr. Gómez-Cruz, 

I'm pleased to inform you that your manuscript has been deemed suitable for publication in PLOS ONE. Congratulations! Your manuscript is now being handed over to our production team.

Kind regards, 

on behalf of

Dr. Yue Gong 

Academic Editor

PLOS ONE